# Weakening of the pinning point buttressing Thwaites Glacier, West Antarctica

Christian T. Wild[1], Karen E. Alley[2], Atsuhiro Muto[3], Martin Truffer[4], Ted A. Scambos[5], and Erin C. Pettit[1]

[1]College of Earth, Ocean, and Atmospheric Sciences, Oregon State University, Corvallis OR, USA
[2]Centre for Earth Observation Science, Department of Environment and Geography, University of Manitoba, Winnipeg MB, Canada
[3]Department of Earth and Environmental Science, Temple University, Philadelphia PA, USA
[4]Geophysical Institute and Department of Physics, University of Alaska Fairbanks, Fairbanks AK, USA
[5]Earth Science and Observation Center, Cooperative Institute for Research in Environmental Sciences, University of Colorado Boulder, Boulder CO, USA

**Correspondence:** Christian T. Wild (wildch@oregonstate.edu)

**Abstract.** The Thwaites Eastern Ice Shelf buttresses a significant portion of Thwaites Glacier through contact with a pinning point 40 km offshore of the present grounding line. Predicting future rates of Thwaites Glacier's contribution to sea-level rise depends on the evolution of this pinning point and the resultant change in the ice-shelf stress field since the break-up of the Thwaites Western Glacier Tongue in 2009. Here we use Landsat-8 feature tracking of ice velocity in combination with ice-sheet model perturbation experiments to show how past changes in flow velocity have been governed in large part by changes in lateral shear and pinning point interactions with the Thwaites Western Glacier Tongue. We then use recent satellite altimetry data from ICESat-2 to show that Thwaites Glacier's grounding line has continued to retreat rapidly; in particular, the grounded area of the pinning point is greatly reduced from earlier mappings in 2014, and grounded ice elevations are continuing to decrease. This loss has created two pinned areas with ice flow now funneled between them. If current rates of surface lowering persist, the Thwaites Eastern Ice Shelf will unpin from the seafloor in less than a decade, despite our finding from airborne radar data that the seafloor underneath the pinning point is about 200 m shallower than previously reported. Advection of relatively thin and mechanically damaged ice onto the remaining portions of the pinning point and feedback mechanisms involving basal melting may further accelerate the unpinning. As a result, ice discharge will likely increase up to 10 % along a 45 km stretch of the grounding line that is currently buttressed by the Thwaites Eastern Ice Shelf.

## 1 Introduction

The Amundsen Sea Coast of West Antarctica has been the setting of drastic glacier retreat, ice-shelf thinning and associated speed-up since the beginning of the satellite measurement era (Scambos et al., 2017). Ice discharge across its grounding line – where ice becomes afloat on the ocean to form ice shelves and glacier tongues – is producing the majority of Antarctica's contribution to contemporary sea-level rise (Rignot et al., 2008; Shepherd et al., 2012). Wherever ice shelves reground on elevated seafloor such as offshore ridges and islands, significant resistance against the flow of floating ice controls the overall

ice-shelf buttressing which, in turn, stabilizes the location of the grounding line (e.g., Alley et al., 2015). This is supported by model experiments showing that reduced ice-shelf buttressing is linked to retreat of grounding lines even far upstream of the location where the change in buttressing occurred, leading to an overall net loss of ice volume above flotation (Dupont and Alley, 2005). Ungrounding of ice shelves from these pinning points, therefore, has the potential to initiate rapid changes on a regional scale (Goldberg et al., 2009; Favier et al., 2012; Favier and Pattyn, 2015; Favier et al., 2016; Reese et al., 2018).

Pinning points exist all around Antarctica's coastline and are classified as either ice rises or ice rumples, depending on their ice-dynamical characteristics (MacAyeal et al., 1987; Matsuoka et al., 2015). Ice rises feature negligible surface motion and divert the general ice flow around them. Prominent examples include Crary Ice Rise on the Ross Ice Shelf, Hemmen Ice Rise on the Ronne-Filchner Ice Shelf, and Bawden Ice Rise on the Larsen C Ice Shelf. Ice rumples, in contrast, maintain the general flow direction and are typically elevated only a few meters above the surrounding ice-shelf surface such as the Doake Ice Rumple enclosed within the Ronne-Filchner Ice Shelf (Johnson and Smith, 1997). The common characteristics of both ice rises and rumples is that they remain grounded throughout the tidal cycle and thus permanently resist the ice-shelf flow through shear stresses at their base. Depending on their formation, these features are categorized as (1) long-term stable, if an ice rise persisted over the last glacial cycle, (2) deglacial emergent, if ice remained locally grounded during ice-sheet retreat, (3) emergent from glacial isostatic adjustment, forming when floating ice regrounds due to post-glacial rebound of the Earth's crust, and (4) glaciological emergent, if thicker ice is advected onto elevated seafloor (Matsuoka et al., 2015).

During the Holocene deglaciation of Antarctica, Thwaites Glacier was grounded on a prominent seafloor ridge ranging between 300-700 m below present-day sea level. The exact timing of ungrounding, causing rapid retreat of about 40 km across a major marine channel (>1.5 km deep) to its present day grounding line, can only be estimated between 40 and 5000 years ago (Tinto and Bell, 2011). Until recently, the main grounding line has been relatively stable on the same seafloor ridge (Fig. 1). Both the slow flowing Thwaites Eastern Ice Shelf and the much faster flowing Thwaites Western Glacier Tongue rested on deglacial emergent ice rises formed at the former grounding-line location (Tinto and Bell, 2011). This relative stability changed drastically by 2009, when the Western Glacier Tongue largely detached from its pinning point and rapidly disaggregated within the following years (Tinto and Bell, 2011; MacGregor et al., 2012; Miles et al., 2020).

Ephemeral grounding close to the 2011 grounding line was detected (Milillo et al., 2019), which indicates the presence of several local seafloor highs, but grounding at low tides generally provides little buttressing to ice flow (Schmeltz et al., 2001). The loss of contact with these small partially grounded seafloor highs, however, allowed the rapid formation of relatively large sub-glacial cavities in seafloor lows behind the initial grounding line, areas that were once shielded against the intrusion of warm modified Circumpolar Deep Water. An example is the 'butterfly' region near the eastern end of Thwaites Glacier's grounding line, where grounding-line retreat rates doubled from 0.6 km $yr^{-1}$ between 1992 to 2011 to 1.2 km $yr^{-1}$ between 2011 to 2017 (Milillo et al., 2019). At the western end of Thwaites Glacier's grounding line, high rates of surface lowering (3-7 m $yr^{-1}$) are attributed to dynamic thinning since the break-up of the Western Glacier Tongue in 2009. Here, the grounding line retreated 14 km between 2009 to 2017 along a deep seafloor trough underlying an embayment in the grounding line where Thwaites Glacier is moving the fastest (>3 km $yr^{-1}$, Fig. 1 c and d). The remaining Eastern Ice Shelf is still confined by an elongated ice rise that is oriented perpendicular to ice flow with visible surface crevassing (photograph in Fig. S1). This

pinning point buttresses the Eastern Ice Shelf and thus a 45 km stretch of the eastern part of Thwaites Glacier's grounding line (more than one third of the full glacier width), but also provides an important control on the pathways for warm modified Circumpolar Deep Water intrusion into the sub-ice-shelf cavity (Wåhlin et al., 2021). The areal extent of the grounded ice on the pinning point not only shrank to half between 1992 and 2011, but the pinning point also split into a larger eastern and a much smaller western portion (Rignot et al., 2014). Further unpinning from its last anchoring points may cause a Western Glacier Tongue-like break-up of the Eastern Ice Shelf in the near future.

For these reasons it is crucial to assess the structural integrity of the pinning point as well as understand signs of its possible destabilization. Here we evaluate its stability through the integration of recent bathymetry estimates and airborne radar surveys from Operation IceBridge and the NERC/NSF International Thwaites Glacier Collaboration (ITGC) with new surface elevation measurements from ICESat-2 satellite laser altimetry. The former is necessary to assess how well the ice-shelf is locally pinned to the seafloor. The latter allows us to derive surface-lowering rates to predict the timing of unpinning and when this 45 km stretch of Thwaites Glacier's grounding line will become fully unconstrained by an ice shelf.

Using airborne radar data, we first modify the existing bathymetry model to match measured ice thickness across the eastern portion of the pinning point. To investigate changes in the regional stability of the Eastern Ice Shelf as well as its pinning point, we compute changes in height above flotation from the Reference Elevation Model of Antarctica (REMA, Howat et al., 2019) and more recent ICESat-2 satellite altimetry data (Smith et al., 2019). We then perform a series of ice-sheet model perturbation experiments to explain the dominant mechanisms behind the observed changes in regional ice-flow dynamics. Lastly, we estimate when Thwaites pinning point will become entirely ungrounded from the seafloor, if the current rates of surface lowering persist. We validate the results with ground-truth data collected in the 2019/20 Antarctic field season as part of the ITGC's Thwaites-Amundsen Regional Survey and Network Integrating Atmosphere-Ice-Ocean Processes (TARSAN) project. We then discuss analogies to the break-up of the Western Glacier Tongue and close by drawing conclusions about the role of the pinning point on tele-buttressing large parts of Thwaites Glacier.

## 2 Data and Methods

### 2.1 Bathymetry adjustment to airborne radar data

Sub-ice-shelf bathymetry of the Eastern Ice Shelf and other ice shelves in the Amundsen Sea was modeled through inversion of airborne gravity data by Jordan et al. (2020). While gravity data can be collected efficiently on a regional scale, they have relatively low spatial resolution (5 km) so only topographic features with wavelengths down to about 5 km are well resolved (Jordan et al., 2020). With a width of about 15 km, the general pattern of Thwaites pinning point is therefore resolved reasonably well, but absolute values remain uncertain (100 m standard deviation). Where ice is grounded on the seafloor, however, ice thickness measurements from airborne radar surveys can be used to further constrain the gravity-derived bathymetry. We compare two data products to the airborne radar data, the 1000 m resolution bathymetry from Jordan et al. (2020) and the 500 m BedMachine v2 bed-elevation product from Morlighem (2020), which was calculated from the mass conservation method where ice is grounded and gravity inversion where ice is floating (Morlighem et al., 2020).

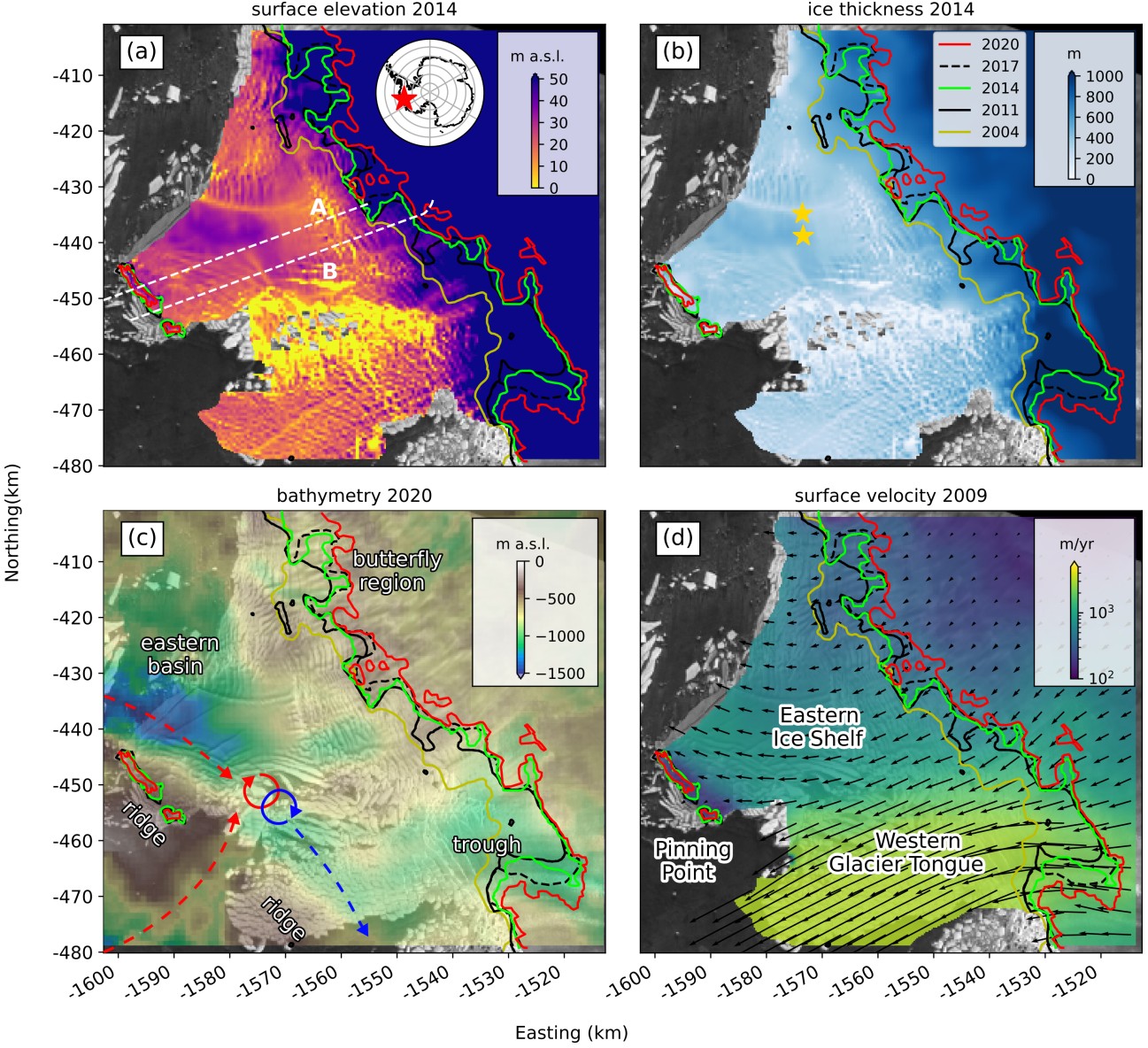

**Figure 1.** Data sets assembled for the Eastern Ice Shelf and the Western Glacier Tongue study area overlain on a Sentinel 1 SAR image from October 2019: (a) Reference Elevation Model of Antarctica (REMA, Howat et al., 2019); (b) BedMachine version 2 ice thickness (Morlighem, 2020); (c) gravity-derived bathymetry (Jordan et al., 2020); (d) NASA's Making Earth System Data Records for Use in Research Environments ice surface velocity product before break-up of the Western Glacier Tongue (MEaSUREs, Rignot et al., 2017). Past grounding lines: yellow is approximately 2004 (Bindschadler et al., 2008); black is 2011 from MEaSUREs, dashed black is 2017 from InSAR (Milillo et al., 2019). We define the green (2014) and red (2020) grounding lines from REMA and ICESat-2 data, respectively. Airborne radar surveys from 2009 and 2019 (panel a) are shown as white dashed lines (radargrams A and B in Fig. 2). The red star in the inset in panel (a) marks the location of Thwaites Glacier in the Amundsen Sea Embayment. The two yellow stars in panel (b) mark the locations of GPS and barometric pressure records used for tidal analysis (AMIGOS stations). The red and blue arrows in panel (c) indicate the pathways of warm Circumpolar Deep Water and cold ice-shelf melt water (Wåhlin et al., 2021). Coordinates in an Antarctic polar stereographic projection (EPSG:3031).

The airborne radar data we used were collected using the Multichannel Coherent Radar Depth Sounder (MCoRDS, Paden et al., 2010, updated 2018) in 2009, and a 600-900 MHz accumulation radar in 2019 (CReSIS, 2020, Fig. 1 a). The ice base was picked from a series of synthetic aperture radar-processed radargrams in a semi-automated fashion. Where ice is grounded on the seafloor, a change in the strength of the basal radar reflection occurs (Schroeder et al., 2016). From the surveys on Thwaites Glacier, only two transects cross the edges of the pinning point in 2009 and only one crosses its central part in 2019. Figure 2 shows two airborne radar transects that reveal a heavily crevassed ice base, suggesting that the Eastern Ice Shelf is far from reaching an equilibrium with the underlying ocean system. If basal melting underneath the floating ice were homogeneous, the ice geometry would be expected to be more uniform (Jordan et al., 2020).

We compare ice thickness from airborne radar measurements over the pinning point to two bathymetry data sets: gravity-derived bathymetry for the Thwaites ice shelves of Jordan et al. (2020) and BedMachine Antarctica Version 2 (Morlighem, 2020). Directly subtracting these bathymetry estimates from the height of the ice base across the pinning point results in average differences of 200±49 m and -150±82 m, respectively. We attribute these differences to the original gravity inversions not being constrained with the radar ice thickness at this isolated location. We therefore fit a quadratic plane to these differences and vertically adjust the bathymetry where ice is grounded according to the 2011 grounding line. As a result of this adjustment, the mismatches between estimated and observed bathymetry are reduced to 0±13 and -74±121 m, respectively (Fig. S2). Because of the large residual error in the BedMachine bathymetry for the eastern portion of the pinning point, we only use the adjusted bathymetry map that is based on Jordan et al. (2020) for further calculations outlined below.

## 2.2 Tidal corrections

Ocean tides as well as the elastic deformation of the Earth's crust underneath the moving water masses (referred to as load tides) influence ice-shelf surface elevation. The predictions of two tide models were validated with available GPS records from the Eastern Ice Shelf: the regional barotropic Circum-Antarctic Tidal Solution (CATS2008) model developed by Padman et al. (2008) and the fully global barotropic assimilation model (TPXO9) from Oregon State University developed by Egbert and Erofeeva (2002). Tidal loading is accounted for in TPXO9, and was added to the CATS2008 model predictions. In addition to the tidal oscillation underneath the floating ice, a 1 hPa increase of barometric pressure on the ice surface causes an isostatic -1 cm response of the ice shelf (Padman et al., 2003). To correct for this inverse barometric effect, we used barometric pressure measured by an automatic weather station (AWS) on Thurston Island (approx. 500 km away across the Amundsen Sea). We validate this record with separate barometric pressure measured on the Eastern Ice Shelf over a 176-day period and find a very good correlation between the measured anomalies ($R^2$=0.94). We note that the Thurston Island AWS also correlates well ($R^2$=0.86) with a much closer AWS located on Bear Island (approx. 200 km away) between 2012 to 2019, before the latter stopped reporting data. The inclusion of the inverse barometric effect does not improve the fit to the GPS data (Fig. S3), suggesting that storm surges and local wind-forced effects on small-basin geometry could be significant (Padman, 2021). The resulting tide correction generally underestimates the vertical displacement measured at the Eastern Ice Shelf within ±17 cm, which is relatively poor compared to the Antarctic-wide accuracy of tide models in coastal areas of about ±10 cm (Padman et al., 2002). The inaccuracy of the tidal correction translates to about ±3 cm yr$^{-1}$ in terms of the derived surface lowering

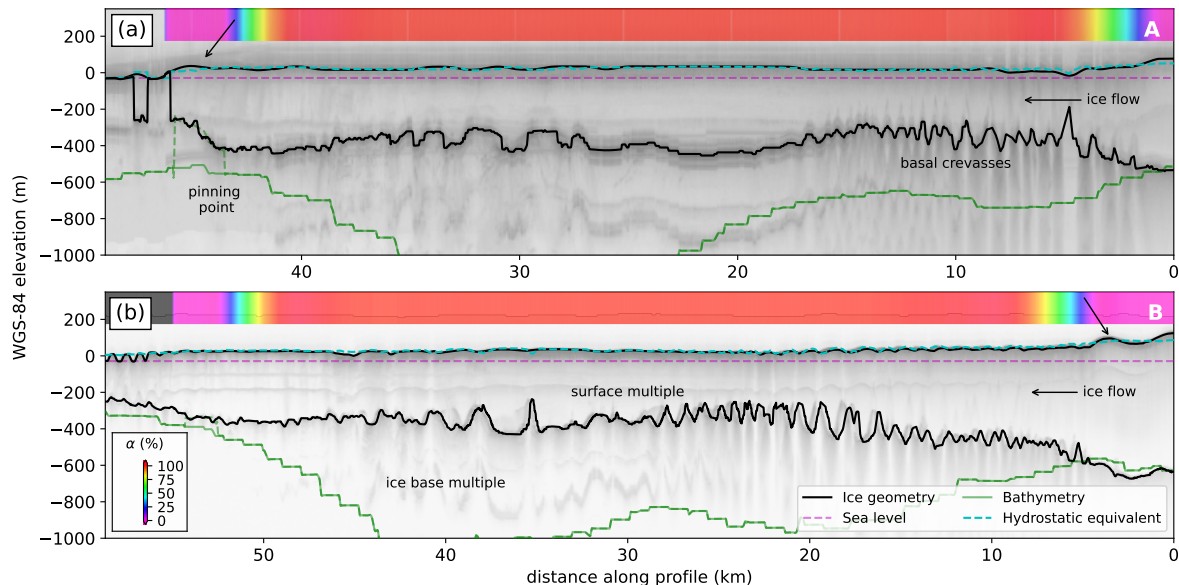

**Figure 2.** Radargrams from airborne radar surveys (a) in Nov 2009 and (b) in Jan 2019 revealing a heavily crevassed ice-base topography. Transects from the grounded ice (right), across the tidal flexure zone as predicted by an elastic model based on the 2011 grounding line (Rignot et al., 2016) onto large parts of the freely-floating ice shelf and (left) ultimately crossing the pinning point that is buttressing the Eastern Ice Shelf. Colors show the percentage tidal displacement with red areas experiencing the full range of the tidal oscillation, while the remaining areas are subject to tidal ice-shelf flexure. Purple areas are grounded according to the 2011 grounding line for the pinning point and the 2017 main grounding line of Thwaites Glacier. The dashed blue line shows flotation height. Grounded areas with surface elevation above flotation height are indicated by black arrows.

rates between 2014 and 2020. We attribute this inaccuracy to either inaccurate tidal constituents in both tide models, errors in the bathymetry used by the tide models, grounding-line location and inaccurate ice-shelf thickness, as well as to insufficient

knowledge of the ice-water drag coefficient (Padman et al., 2018).

Along a narrow band oceanward of the grounding line, ice is not freely floating because longitudinal stresses within the ice limit ice-shelf flexure. We numerically model the vertical displacement due to this tidal flexure with the well-known elastic approximation (Holdsworth, 1969; Vaughan, 1995; Schmeltz et al., 2002; Sayag and Worster, 2013) as formulated by Walker et al. (2013):

$$k\mathbf{w} + \nabla^2\left(\mathbf{D}\nabla^2\mathbf{w}\right) = \mathbf{q},$$  (1)

$$\mathbf{D} = \frac{E\mathbf{H}^3}{12(1-\lambda^2)},$$  (2)

$$\mathbf{q} = \rho_{sw} g \left( A - \mathbf{w} \right), \tag{3}$$

where $\mathbf{w}$ is elastic vertical deflection, $\nabla^2$ the 2D Laplace operator and $k$ a spring constant of the foundation which is zero for the floating part. $\mathbf{D}$ is vertically integrated ice-shelf stiffness while undergoing tidal bending (Love, 1906, p. 443) with $E = 1.5$ GPa the effective Young's modulus, $\mathbf{H}$, is ice thickness given by BedMachine and $\lambda$ is Poisson's ratio of a Maxwell model (Gudmundsson, 2011) accounting for transverse deformation due to longitudinal strain. The tidal force underneath the floating ice, $\mathbf{q}$, is given by the tidal amplitude $A = 1$ m, the density of ocean water $\rho_{sw}$ and gravitational acceleration $g$. Symbols used in this study are summarized in Table A1. For boundary conditions, we implement a fulcrum at the grounding line ($\mathbf{w} = 0$) and anchor the upstream boundaries of the model domain of the grounded portion rigidly ($\mathbf{w} = 0, \nabla^2 \mathbf{w} = 0$). Generally, the elastic flexure pattern is time-invariant and only dependent on ice-shelf stiffness (Wild et al., 2017). We therefore normalize the finite-element solution for the deflection, $\mathbf{w}$, with the applied tidal forcing to derive tide-deflection ratio throughout the area ($\alpha$-map, Han and Lee, 2014, Fig. S4). By extracting the $\alpha$-value for a given location, tide model output from freely-floating parts of the ice shelf can thus be directly scaled to include the effects of 2D elastic tidal flexure near grounding lines (Wild et al., 2019). The computationally expensive task of running the finite-element model over time is thus circumvented. By using the tide-deflection ratio, inaccuracies in the $\alpha$-map due to a possibly incorrect ice-shelf stiffness are always smaller than inaccuracies in the utilized tidal forcing. We estimate these within the reported uncertainty of the effective Young's modulus from tiltmeter data elsewhere ($\pm 0.69$ GPa, Wild et al., 2017) to a maximum of $\pm 0.08$ m. In areas where ice is grounded, $\alpha$-values close to or equal zero prevent the application of any tidal correction of GPS and ICESat-2 measurements.

## 2.3 Height above flotation calculation

We use height above flotation as a proxy to assess changes in the degree of grounding. Ice-surface elevations are given by the continent-wide and time-stamped REMA data set (Howat et al., 2019). REMA data in our study area were mostly acquired in 2014 with feathered strip edges in the vicinity of our region of interest where we expect the largest inaccuracies. To detect ice-surface elevation change since the acquisition of REMA, we use the ICESat-2 L3A Land Ice Height, Version 3, data set (Smith et al., 2019), which gives absolute ice-surface elevations with under 3 cm vertical and 9 cm horizontal accuracies (Brunt et al., 2019). We removed about 16 % of the ICESat-2 points with the provided quality summary flag. ICESat-2 data in our study region were acquired in 2019/20 and allow us to calculate changes in ice-surface elevation since 2014 when compared to REMA.

We first translate ice-shelf surface elevations from both REMA and ICESat-2 data to freeboard by using EIGEN6c4 geoid model (Förste et al., 2014) as the mean sea level. We neglect the effect of mean dynamic topography as it is not directly measurable on the ice-shelf surface and invert freeboard, $\mathbf{z_f}$, to floating ice thickness, $\mathbf{H_f}$, based on hydrostatic equilibrium principles. Here, we take advantage of the two available airborne radar transects and use all traces acquired on the freely-floating part of the Eastern Ice Shelf to derive a single mean ice-column density ($\bar{\rho}$) for this inversion from each of the radargrams ($893 \pm 21$ kg m$^{-3}$ for 2009 and $871 \pm 10$ kg m$^{-3}$ for 2019/20, respectively). We attribute the decline in mean ice-column density

of the Eastern Ice Shelf to the interplay of a prolonged erosion of dense ice at the ice-shelf base where $\rho_{ice} = 917 \, \text{kg m}^{-3}$, with an increase of relatively soft snow accumulation over the last decade on the surface where $\rho_{firn} = 430 \pm 66 \, \text{kg m}^{-3}$ from snow-pit measurements. Modelled surface mass balance further supports our observation of a decline in mean ice-column density (Keenan et al., 2021). This trend is independently confirmed by comparing REMA surface elevations and BedMachine ice thickness data from 2014 to field measurements on 38 sites using GPS and phase-sensitive radar in the 2019/20 season ($883 \pm 7$ and $865 \pm 12 \, \text{kg m}^{-3}$, respectively). On grounded areas including the pinning point, we then directly subtract the adjusted seafloor depths from REMA and ICESat-2 surface elevations to calculate absolute ice thickness, $\mathbf{H_a}$, in 2014 and 2019/20, respectively. Lastly, the difference between absolute and our hydrostatic ice thickness estimate is calculated and expressed as height above flotation, $\mathbf{z_f}$:

$$\mathbf{z_f} = (\mathbf{H_f} - \mathbf{H_a}) \frac{\rho_{sw} - \bar{\rho}}{\rho_{sw}}. \tag{4}$$

We then use the transition between grounded areas that always feature a positive height above flotation to floating areas ($\mathbf{z_f} = 0$ m a.f.), to manually delineate the location of the grounding line in 2014 and in 2019/20. These estimates did not prove sensitive to the reduced value of mean ice-column density beyond the inaccuracies arising from manually picking the grounding line. We estimate this uncertainty to about $\pm 0.1 \, \text{km yr}^{-1}$ in terms of grounding-line retreat rates. We calculate retreat rates from comparison with the 2004 grounding line from Bindschadler et al. (2011), which uses 1999-2003 Landsat and 2003-2009 ICESat-1 data, the InSAR-derived 2017 grounding lines from Milillo et al. (2019) and, where these are unavailable, such as at the pinning point, with the 2011 grounding line from Rignot et al. (2016).

### 2.4 Lagrangian analysis of surface lowering

A Lagrangian framework tracks ice parcels with time and thus corrects for the effects of ice-flow advection during change calculation. This is necessary as the uneven ice base of the Eastern Ice Shelf (Fig. 2) combined with relatively fast ice flow induces an advection signal that may spuriously be misinterpreted as rapid surface-height change if left uncorrected for. We use the annual velocity field composite from Landsat-8 feature tracking between 2013 and 2020 (Alley et al., 2021) and migrate the ICESat-2 point cloud back in time to where each point would have been when REMA data were acquired. We therefore capture ice-dynamical changes such as an observed counter-clockwise rotation of the ice flow and the associated change in ice-flow divergence in the migration. No seasonality of flow-direction nor any inter-annual change in speed were detected in the velocity record. We estimate accuracy by validating our velocity record with GPS measurements at 38 sites to $28 \, \text{m yr}^{-1}$, or 4.5 % of the average flow speed.

We then use the $\boldsymbol{\alpha}$-map in combination with tide modeling to correct for elastic tidal flexure as ICESat-2 points acquired at different times migrate along streamlines across the grounding zone. As a result, rates of surface lowering are valid on both grounded and floating ice. We then rasterized the migrated ICESat-2 point cloud using a 2D Gaussian kernel to obtain an approximately 68-m resolution map of surface-lowering rates since 2014, when REMA data were acquired. Overall accuracy of these rates is estimated to be the squared sums of errors from REMA, ICESat-2, the applied tidal correction and the accuracy of our velocity record, divided by the 5-6 years of time difference to ICESat-2 data acquisition. This conservative estimate results

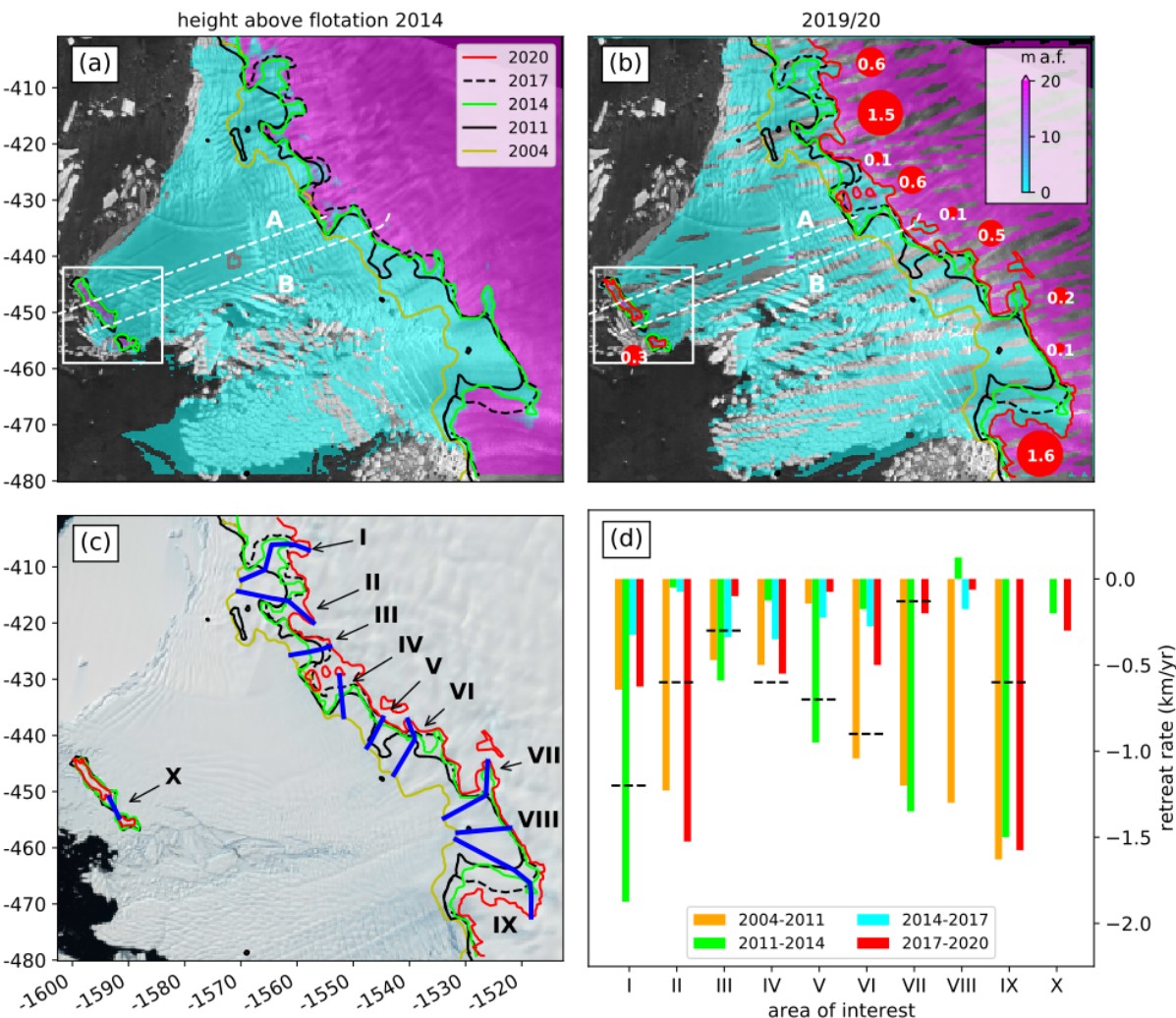

**Figure 3.** Regional changes: Height above flotation from (a) REMA data in 2014 and (b) ICESat-2 surface altimetry data in 2019/20 overlain on a Sentinel 1 SAR image from October 2019. Airborne surveys from 2009 and 2019 (panels a and b) are shown as white dashed lines. Red circles in panel (b) show rates of grounding-line retreat since the last assessment in 2017 (Milillo et al., 2019). Roman numerals in panel (c) refer to areas of interest discussed in the text and are overlain on the Landsat Image Mosaic of Antarctica (Bindschadler et al., 2008). Panel (d) shows histograms of grounding-line retreat rates along the (blue) profiles in panel (c) in comparison to (dashed black) the InSAR-derived retreat rates between 2011 to 2017 from Milillo et al. (2019). The white rectangle (panels a and b) shows the spatial extent of Figure 5.

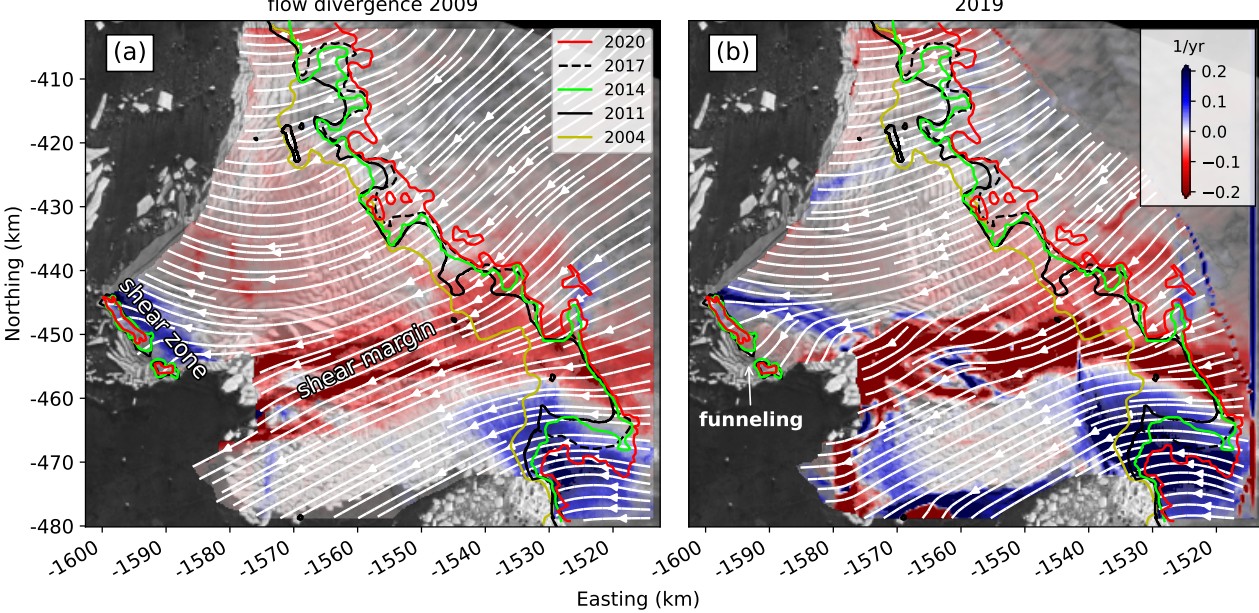

**Figure 4.** Ice flow divergence from (a) MEaSUREs data in 2009 and (b) Landsat-8 feature tracking in 2019 overlain on a Sentinel 1 SAR image from October 2019. Blue colors denote flow divergence, red colors flow convergence. Note the upstream migration of the shear zone near the pinning point and the funneling of ice streamlines between the two remaining portions of the pinning point.

in an accuracy of the derived surface lowering-rates of $\pm 1.25$ m yr$^{-1}$ at a maximum, which is well below the detected change signal. We note that an area-wide uncertainty of about 6 m in REMA is the largest contribution to this accuracy estimate and originates mainly from a narrow area where REMA strip edges have been feathered to form a seamless mosaic. This narrow area lies largely outside of our region of interest (Fig. 6 b), which increases our confidence in accuracy within the boundaries of individual REMA strips.

## 2.5 Ice-dynamics modeling

We use a widely-used ice-sheet model to investigate causes of the observed changes in ice dynamics. If the recent and present changes can be replicated well, numerical modeling can be used to provide insights into likely scenarios of future change and sensitivity to various forcings. The Ice-Sheet and Sea-level System Model (ISSM, Larour et al., 2012) is a thermo-mechanically coupled model capable of calculating higher-order stress components of ice dynamics that relies on the classical conservation laws of physics such as the balance of stresses and the incompressibility of ice. Deforming ice is represented in the model equations with a purely viscous, non-linear rheology following the Nye-Glen isotropic relation (Nye, 1953; Glen, 1955):

$$\boldsymbol{\sigma}_{ij} = \frac{1}{2} \mathbf{B} \dot{\epsilon}_e^{\frac{1}{n}-1} \dot{\boldsymbol{\epsilon}}_{ij}, \tag{5}$$

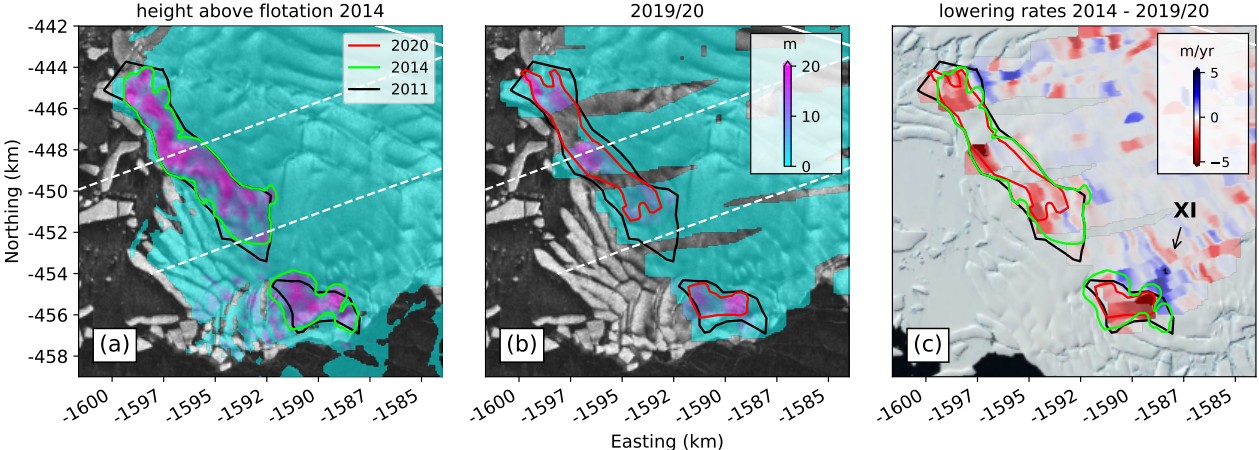

**Figure 5.** Close-up of the pinning point area and calculated height above flotation (a) from REMA and (b) ICESat-2 data. (c) Lagrangian rates of surface lowering. Red colours indicate thinning, while blue areas delineate thickening. The green and red lines show the reduced extent of the pinning point as estimated from height above flotation in 2014 and 2019/20, respectively. Map background is a Sentinel 1 SAR image from October 2019, and the Landsat Image Mosaic of Antarctica (Bindschadler et al., 2008). The Roman numeral in panel (c) refers to an area of interest discussed in the text.

where $\boldsymbol{\sigma}_{ij}$ is the deviatoric stress tensor, $\dot{\epsilon}_e$ the effective strain rate of deformation defined as the second invariant of the strain rate tensor, $\dot{\boldsymbol{\epsilon}}_{ij}$. Our model simulations employ a depth-averaged rheological parameter, $\mathbf{B}$, with a creep exponent of $n = 3$. Rheology is dependent on ice temperature and crystal fabric, which are both not explicitly implemented in the model, so fields for $\mathbf{B}$ are inferred from surface velocity data. Higher values of $\mathbf{B}$ indicate colder/stiffer ice, whereas lower values indicate warmer/damaged ice. Where ice is grounded, friction at the ice base exerts a drag force, $\boldsymbol{\tau_b}$, opposing the ice flow that is directly proportional to the basal velocity, $\boldsymbol{v_b}$ (Cuffey and Paterson, 2010). We use a linear, Budd-type friction law (Budd et al., 1984):

$$\boldsymbol{\tau_b} = -\boldsymbol{\beta}^2 \mathbf{N} \boldsymbol{v_b}, \tag{6}$$

where $\mathbf{N}$ is the effective water pressure at the ice base given by $\mathbf{N} = g(\rho_{ice}\mathbf{H} + \rho_{sw}\mathbf{z_b})$, which becomes zero for floating ice to eliminate vertical gradients of horizontal velocities within ice shelves. This formulation assumes a perfect hydrological connection between the ocean and the grounded ice base and neglects dynamic subglacial water flow, such as lake drainage or basal discharge (Wingham et al., 2006; Fricker and Scambos, 2009). The basal friction coefficient, $\beta^2$, cannot be measured directly. On grounded ice, positive values of $\beta^2$ enable horizontal shear within the ice column, which can be used to infer properties of the bed itself by comparing observed and modeled velocity fields on the ice surface. Here we use the Shallow Shelf (or Shelfy-Stream) Approximation (SSA, Morland, 1987; MacAyeal, 1989) to infer both $\mathbf{B}$ and $\beta^2$ in a series of inverse

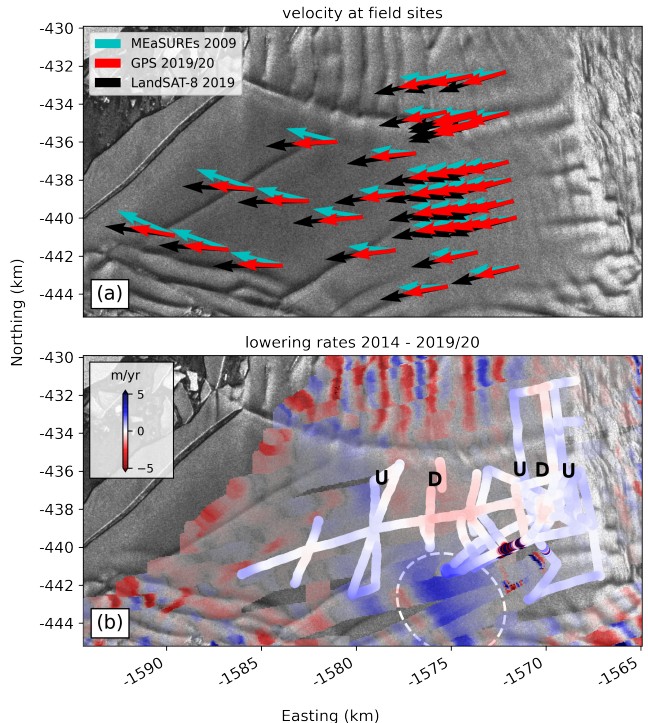

**Figure 6.** Validation of remote sensing observations with static and kinematic GPS measurements on the Eastern Ice Shelf overlain on a Sentinel 1 SAR image from October 2019: (a) Counter-clockwise rotation of ice flow between 2009 from MEaSUREs to 2019 from Landsat-8 feature tracking in comparison with results from a recent GPS repeat survey. The black Landsat-8 vectors have been lengthened by 33 % to increase visibility under the red GPS vectors and are in excellent agreement with the measurements. (b) Comparison between recent Lagrangian rates of surface lowering from a kinematic GPS survey to validate Lagrangian surface-lowering rates derived from ICESat-2 data. Positive values indicate thickening, while negative-value areas delineate vertical thinning. White colours were coded transparent in the ICESat-2 data to highlight the GPS tracks. Tracks were plotted at their migrated locations in 2014. The letters U and D indicate areas of local hydrostatic imbalance, where the ice surface is going up and down, respectively. The dashed white oval shows the location of a spurious thickening signal that we interpret as an artefact from feathering between individual REMA strips.

steps by minimizing the mismatch between observed and modeled velocity fields (Appendix A and Fig. S5). In the SSA, horizontal velocity does not vary with depth in the ice column and vertical gradients in shear stress are hence neglected. As a consequence, the derived fields of **B** are vertically-integrated and $\beta^2$ is directly derived from the surface velocity. We then solve the stress balance using the SSA in a series of diagnostic model runs. The finite-element mesh is optimized using static anisotropic adaptation techniques to provide fine resolution near grounding lines and narrow shear zones as guided by surface velocity fields during model initialisation (Larour et al., 2012). We use a Bidimensional Anisotropic Mesh Generator (Hecht, 1998) to create a final mesh consisting of 44168 unstructured elements with a minimum edge length of 100 m and a maximum edge length of 450 m (Fig. S6).

Two different velocity data sets are compared in our modeling work to invert for $\mathbf{B}$ over the entire domain and $\beta^2$ over grounded ice. To compute an unperturbed reference state, we use the Making Earth System Data Records for Use in Research Environments product (MEaSUREs, Rignot et al., 2017), which originates from interferometric synthetic aperture radar data acquired between 2007 and 2009 when the Western Glacier Tongue was largely intact. We compare MEaSUREs with a recent velocity field in 2019, which we derived from Landsat-8 imagery collected after the break-up of the Western Glacier Tongue. Bed elevation and ice thickness are prescribed by our adjusted bathymetry and the REMA surface elevation. We outline the model domain guided by the drainage basins feeding into the floating parts of Thwaites Glacier and the location of the ice front. For the perturbation experiments, we first delineate the domain using visible satellite imagery pre- and post-break-up of the Western Glacier Tongue, and later modify the grounded mask at the pinning point as guided by our analysis of height above flotation. We impose a Dirichlet boundary condition ($\boldsymbol{v} = \boldsymbol{v}_{obs}$) along the upstream limit of the domain using observed velocity and a Neumann stress boundary condition ($\boldsymbol{\sigma} \cdot \boldsymbol{n} = -p_{sw}\boldsymbol{n}$) along the ice-shelf front prescribed by the hydrostatic pressure $p_{sw} = -\rho_{sw}gz$ of the water column (Fig. S6). The ice surface is assumed stress-free. We note that the diagnostic model does not solve for the temporal evolution of the ice geometry as a response to the applied environmental perturbations, but proves valuable to identify the main drivers behind the observed ice-dynamical changes between the two velocity fields by solving the instantaneous stress balance.

To investigate the effects of the break-up of the Western Glacier Tongue and a vanishing pinning point on the regional ice-flow configuration we perform a series of perturbation experiments (Tab. 1) to detect the underlying mechanisms behind the observed changes in ice dynamics: (i) counter-clockwise rotation of the ice flow since break-up of the Western Glacier Tongue; and (ii) ice funneling through the two remaining portions of the pinning point. To address (i), we run the model with and without the full extent of the Western Glacier Tongue (Fig. 7). Then, (ii) is addressed by reducing the spatial extent of the pinning point as guided by our analysis of height above flotation and comparison with model solutions when the pinning point has its full 2011 extent. Finally, we entirely unpin the Eastern Ice Shelf from the pinning point and recalculate the stress balance to model the regional response of the ice-shelf system on the predicted ungrounding. All other ISSM simulations use the 2011 location of the grounding line.

**Table 1.** Setup of the perturbation experiments in ISSM

| Simulation | Western Glacier Tongue | Pinning Point | Velocity field | Description |
|:---:|:---:|:---:|:---:|---:|
| (a) | Yes | 2011 | MEaSUREs 2009 | Control run MEaSUREs |
| (b) | No | 2011 | MEaSUREs 2009 | break-up of the Western Glacier Tongue |
| (c) | Yes | 2011 | Landsat-8 2019 | Control run LandSAT-8 |
| (d) | No | 2011 | Landsat-8 2019 | break-up of the Western Glacier Tongue |
| (e) | Yes | 2019/20 | Landsat-8 2019 | Weakening of the pinning point from ICESat-2 |
| (f) | Yes | ungrounded | Landsat-8 2019 | Removal of the pinning point |

## 3   Results

### 3.1   Regional grounding-line retreat

Integrating data from airborne radar surveys with the REMA digital elevation model and recent ICESat-2 measurements reveals temporal changes in height above flotation from 2014 to 2020. Grounding-line retreat of Thwaites Glacier is known to be heterogeneous with retreat rates averaging to about 0.8 km yr$^{-1}$ and peaking up to 1.2 km yr$^{-1}$ between 2011 and 2017 along preferential channels in the bathymetry since the last assessment (Milillo et al., 2019). Our results show continued grounding-line retreat along a 45-km stretch of Thwaites Glacier's grounding line, with sporadically accelerating retreat rates.

While a number of new ice rises emerged during recent deglaciation on bathymetric bumps, several new cavities are forming or have already opened up to the intrusion of seawater. To highlight this heterogeneity we present ten areas of interest that show different retreat patterns (Fig. 3 c and d): (I) Retreat on the Eastern wing of what is referred to as the 'butterfly' region in the literature has slowed down from 1.2 km yr$^{-1}$ between 2011 to 2017 to about 0.6 km yr$^{-1}$ between 2017 to 2020. The maximum inland reach of its western wing remained stable since 2017 as this cavity is enclosed by a well-grounded bathymetric ridge that is currently preventing further retreat. Height above flotation indicates that the majority of the 'butterfly' region was already floating, or at least close to flotation in 2014, suggesting that ungrounding likely occurred between 2011 and 2014 with retreat rates up to 1.9 km yr$^{-1}$. (II) To the West of the 'butterfly' region, a newly formed cavity is rapidly expanding with retreat rates up to 1.5 km yr$^{-1}$ since 2017. Further retreat along this narrow bathymetric low would cut off a newly formed ice rise. However this is unlikely in the near future as the remaining 2.5 km are well grounded with $z_f > 15$ m. (III) Grounding-line retreat rates within this narrow embayment steadily decelerated since 2011, which is in conjunction with the bed elevation rising in the inland direction as observed in the bathymetry data (Fig. 1 c) and matches the 0.3 km yr$^{-1}$ between 2011 to 2017 from Milillo et al. (2019). (IV) A large grounded protrusion at this location has been retreating constantly at an accelerating rate of 0.4-0.6 km yr$^{-1}$ within the central part of the grounding line, and it has now formed three well-grounded ice rises with $z_f > 18$ m. (V) Here, rapid retreat of 1 km yr$^{-1}$ between 2011 to 2014 slowed down to about 0.1 km yr$^{-1}$ due to a bathymetric bump. (VI) Similar to area II, a new ice rise will likely emerge within only a few years as the remaining 1.6 km between the current grounding line and an inland bathymetric depression ungrounds at a rate of 0.5 km yr$^{-1}$. (VII) Grounding-line retreat of up to 0.5 km yr$^{-1}$ is observed compared to the 2017 grounding line along a cavity that stretches into the main trunk of Thwaites Glacier. However, our analysis of height above flotation reveals that much of this cavity was already close to flotation in 2014, which would change this rate to 0.8 km yr$^{-1}$ between 2014 to 2020. Further inland, we detect an isolated area that is already at its level of flotation but is currently shielded from warm water intrusion by a seafloor saddle. (VIII) Here, the grounding line remained almost stable since 2011 due to the locally elevated seafloor that is separating the eastern from the western branch of Thwaites Glacier, feeding into the former Western Glacier Tongue. (IX) Accelerated grounding-line retreat up to 1.6 km yr$^{-1}$ along this deep seafloor trough where Thwaites Glacier flows the fastest. (X) Further offshore, where the Eastern Ice Shelf is confined by the pinning point, the two remaining eastern and western portions are separating at an accelerated rate of 0.3 km yr$^{-1}$ between 2014 to 2020, compared to 0.2 km yr$^{-1}$ between 2011 to 2014. This opening now

allows the ice to funnel over a bathymetric saddle in between the portions of the pinning point. The temporal evolution of the observed grounding-line retreat rates is summarized in Figure 3d.

## 3.2 Changes in ice-dynamics

Here we summarize the relevant processes that directly impact stability of the pinning point. An in-depth study on the history of dynamical changes at both the Eastern Ice Shelf and Western Glacier Tongue during the satellite measurement era is published in Alley et al. (2021).

We use the two velocity fields and detect significant changes in ice-flow divergence over the last decade (Fig. 4). Before the break-up of the Western Glacier Tongue, ice flowed mainly through the calving front to the east. We interpret this as evidence

of significant buttressing of the Eastern Ice Shelf through the pinning point. In the years following the break-up of the Western Glacier Tongue in 2009, the ice flow on the main floating part of the Eastern Ice Shelf has (i) rotated counter-clockwise, (ii) streamlines are now funneling through a narrow gap between the eastern and western portions of the pinning point (white streamlines in Fig. 4 b and 8 b), and (iii) the shear zone upstream of the pinning point weakened and migrated up to 10 km upstream. Alley et al. (2021) hypothesize that this counter-clockwise rotation is attributed to the break-up of the Western

Glacier Tongue, following years of increased lateral drag.

Along the coast, flow across the eastern half of Thwaites Glacier's grounding line has remained largely unchanged. This indicates the sustained ice-shelf buttressing of this area by the Eastern Ice Shelf. The western half, in turn, exhibits increased absolute values of flow divergence, which is either a result of accelerating ice discharge across the grounding line as the grounded ice is still adjusting to the removal of the Western Glacier Tongue (Figs. 7 and 8) or triggered by reduced ice-shelf

buttressing and subsequent new ice divergence because of shrinking of the pinning point (Fig. 9).

We validate these remote-sensing observations with ground-truth data from a 12-day GPS repeat-survey collected in the 2019/20 season (Fig. 6). GPS data were processed using the Precise Point Positioning kinematic solutions from the Canadian Geodetic Survey (https://webapp.geod.nrcan.gc.ca/geod/tools-outils/ppp.php). Errors were estimated from the standard deviation of the solutions. At 38 sites distributed on the freely floating part of the Eastern Ice Shelf, our GPS measurements

support the counter-clockwise rotation of the ice flow observed with satellite data from $89\pm8°$ in 2009 to $101\pm6°$ in 2019/20 from east with an average flow direction of $99\pm6°$ in the GPS observation. Simultaneously, average flow velocities at these sites decreased from $661\pm79$ m yr$^{-1}$ in 2009 to $597\pm23$ m yr$^{-1}$ in 2019, which fits well with $625\pm28$ m yr$^{-1}$ of the GPS observation. The spatial distribution of these changes shows that the largest counter-clockwise rotation of ice flow occurred just a few kilometers upstream of the pinning point, where the ice flow accelerated since 2009 (Fig. 6 a). Because our velocity

field derived from Landsat-8 feature tracking captures the measured velocity field at these 38 sites within the errors, we trust the validity of satellite-derived streamlines in areas beyond the field survey, such as near the pinning point and the grounding line where a series of rifts and crevasses prevented the collection of further ground-truth data.

### 3.3 Ungrounding of the pinning point

The bathymetry product of Jordan et al. (2020) shows that the elevation of the seafloor ridge on which the pinning point is
located is -498±72 m. Airborne radar data from 2009 and 2019 reveal that underneath the eastern portion of the pinning point,
the ridge is several hundred meters higher at -348±74 m when compared to the BedMap2 dataset from Fretwell et al. (2013).
The average height above flotation of the eastern pinning point is 7.6±6.3 m in 2019/20 indicating that only modest ice-column
thinning is required to detach from the seafloor ridge underneath (Fig. 5).

Rates of surface-height change also reveal a thickening signal upstream of the smaller western portion of the pinning point of
up to 5 m yr$^{-1}$ (area of interest XI in Fig. 5 c). The additional basal shear stress from localized grounding induces compressional
stresses, and thus thickening, upstream. A similar thickening signal, although of smaller magnitude (3 m yr$^{-1}$) is observed
upstream of the eastern edge of the pinning point. Validation of the lowering rates from satellite data with kinematic GPS data
acquired within the area of safe field operations in the 2019/20 field season shows a similar pattern as observed with ICESat-2
(Fig. 6 b). Differences between the derived lowering rates are -0.28±0.66 m yr$^{-1}$ with ICESat-2, slightly underestimating
the GPS measurements. We attribute this small mismatch to moving wind-sculpted snow features (sastrugi) on the ice-shelf
surface. We also identify a series of alternating thickening and thinning signals oblique to ice flow, which we interpret as the
ice shelf's response to heterogeneous basal melt patterns and subsequent small-scale hydrostatic adjustment.

With the derived information about stability and rate of destabilization of the pinning point, it is possible to estimate when the
Eastern Ice Shelf will likely unpin entirely from the seafloor ridge. Average height above flotation on both remaining portions
of the pinning point is about 8.0±6.5m (Fig. 5 b). Assuming that recent surface-lowering rates between 2014 and 2019/20 of
-0.3±1.5 m yr$^{-1}$ persist into the future, unpinning will happen within the next 20 to 30 years. There is, however, the potential
to detach within the next decade if the true height above flotation is on the lower end and the true surface-lowering rates are on
the higher end of the reported range. Since ice-shelf thinning involves a multitude of highly non-linear processes that speed-up
the ungrounding, we hypothesize that entire unpinning within the next decade is very likely.

### 3.4 Numerical modeling of the ice-shelf response

We compare two model configurations with the aim to evaluate the net effect of the weakening shear margin on the influence of
the Western Glacier Tongue. For example, rheologically weaker ice in the shear margin will progressively decouple the Eastern
Ice Shelf from the fast-flowing Western Glacier Tongue. The derived fields of the rheological parameters capture the weakening
of the shear margin between 2009 and 2019 (Fig. A2), which further supports our hypothesis of an ongoing reconfiguration of
regional ice dynamics.

The basal friction coefficients as derived from the two velocity fields show relatively low to moderate values around 10-100
(Pa yr/m)$^{1/2}$ for the majority of the grounded areas (Fig. A2). This is in general agreement with Full-Stokes inverse modeling
efforts from Morlighem et al. (2013), with values between 0-50 (Pa yr/m)$^{1/2}$. Low basal friction coefficients indicate that sliding
over the ice-bed interface is more likely than a no-slip type of internal deformation, when ice is frozen to the bed. Lubricated
subglacial conditions, in turn, increase the radar basal reflectivity, which led to the conclusion that much of Thwaites Glacier's

bed is thawed (Schroeder et al., 2016). Our inferred fields of the basal friction coefficients support this with grounded parts underneath Thwaites Glacier's main trunk showing large patches of slippery subglacial conditions indicated by very low basal friction coefficients (Fig. A2). At the pinning point and isolated bathymetric highs along the 2011 grounding line, we required to significantly increase basal friction coefficients of up to 1000 (Pa yr/m)$^{1/2}$ to match the satellite observed velocity fields.

These high values were necessary to slow down the modeled ice flow. In reality, localized straining and fracturing in this area further lowers the effective ice viscosity, which reduces longitudinal stresses on the pinning point and thus inferred basal drag.

The control run that consists of the MEaSUREs velocity field, a model domain that includes the Western Glacier Tongue, and the full extent of the 2011 pinning point shows that the inversion is successful in finding a basal friction coefficient field that reproduces the observed velocity field (Fig. 8). Velocity misfits are $-2 \pm 100$ m a$^{-1}$ over the entire domain, $1 \pm 43$ m a$^{-1}$ on the

365 Eastern Ice Shelf and $-1 \pm 100$ m a$^{-1}$ on the Western Glacier Tongue (Fig. S5 a). We then remove the Western Glacier Tongue from the computational domain and solve the stress balance again for the modelled velocity field, keeping the basal friction coefficients from the control run. The modeled streamlines are almost perpendicular to the orientation of the pinning point and represent a counter-clockwise rotation not observed in the MEaSUREs velocity field (Fig. S7 b). This experiment suggests that the Western Glacier Tongue was still affecting ice dynamics on the Eastern Ice Shelf when data for the MEaSUREs product

were acquired. However, relative to the field-acquired GPS vectors (Fig. 6 a), the modeled streamlines slightly overestimate the influence of the Western Glacier Tongue.

To further test the hypothesis whether the counter-clockwise rotation of ice flow is a consequence of the break-up of the Western Glacier Tongue we repeat the perturbation of removing the Western Glacier Tongue from the model domain but replace the MEaSUREs velocity field from 2009 with our Landsat-8 record from 2019. The modeled streamlines of the second control

run match the satellite derived streamlines near the pinning point, but slightly over-estimate the counter-clockwise rotation of ice flow on the Eastern Ice Shelf (Fig. S7 c). Velocity misfits are $-10 \pm 282$ m a$^{-1}$ over the entire domain, $-2 \pm 100$ m a$^{-1}$ on the Eastern Ice Shelf and $-10 \pm 282$ m a$^{-1}$ on the Western Glacier Tongue (Fig. S5 b). This mismatch is a consequence of including the Western Glacier Tongue in the model domain, although ice rheology in the shear zone weakened (Fig. A2). Removing the Western Glacier Tongue entirely from the model domain results in modeled streamlines that also over-estimate

the counter-clockwise rotation of ice flow (Fig. 8 b), which confirms the response of the first perturbation experiment. Both simulations suggest that although lateral drag across the shear margin is reduced, the Western Glacier Tongue still affects ice flow on the Eastern Ice Shelf, particularly in areas closer to the grounding line, where closely spaced icebergs to the west still transmit lateral shear stresses.

Finally, we investigate the effect of continued weakening of the pinning point on the regional ice flow pattern (simulations

e and f) by reducing the spatial extent of the pinning point as guided by our analysis of height above flotation from ICESat-2. This requires modifying the grounded mask in ISSM by designating model elements between the eastern and western portion of the pinning point as floating. ISSM then solves the stress balance again diagnostically that is directly compared to the simulation based on the full 2011 extent of the pinning point (simulation c). Differences between the modeled stress balances show increased ice funneling across a narrow bathymetric saddle between the eastern and western portion of the pinning point

(Fig. 9 a) that is also evident in visible satellite imagery (Fig. S8). The induced weakening leads to an increase of ice flow

velocities that reaches from the floating parts of the Eastern Ice Shelf upstream and across the grounding line. The western half of the Eastern Ice Shelf is more affected than its eastern half, with a mean increase in ice flow velocities of $124\pm66$ m yr$^{-1}$ across the grounding line or about $9\pm5$ % of the 2019 velocity from Landsat-8 feature tracking. Between the two remaining portions of the pinning point, ice velocities almost double with the removal of basal shear stress, underlining the funneling effect

between the eastern and western portions of the pinning point. Entire ungrounding from the pinning point (Fig. 9 b) causes a widespread reconfiguration of regional ice dynamics with the Eastern Ice Shelf doubling in speed according to the stress balance. We interpret this as the model's representation of a Western Glacier Tongue-like break-up following the unpinning. Mean ice-flow velocities across the grounding line increase by $154\pm63$ m yr$^{-1}$ corresponding to $10\pm3$ % of the 2019 velocity.

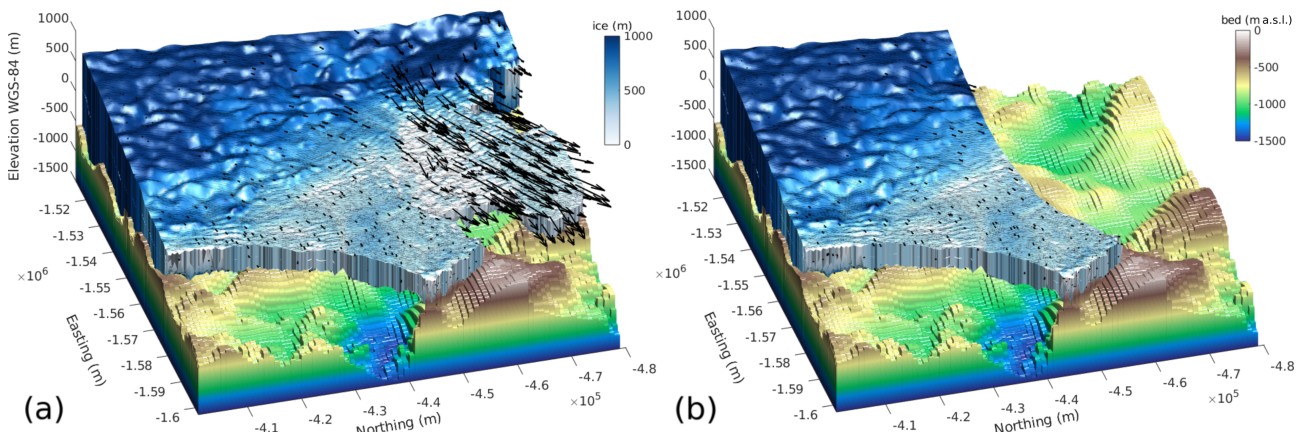

**Figure 7.** ISSM computational domains used for perturbation experiments with and without the Western Glacier Tongue. Note the deep bathymetric trough stretching from the pinning point area far beneath the Western Glacier Tongue. The bathymetry is our adjusted version of the Jordan et al. (2020) product, and ice thickness is derived from hydrostatic principles on floating parts and from differencing the REMA surface elevation with the bathymetry on grounded parts. Note that the Figure is rotated in comparison to all previous Figures.

## 4   Discussion

Our perturbation experiments confirm our hypothesis that weakening of the lateral shear margin in the years following break-up of the Western Glacier Tongue is the mechanism behind the observed counter-clockwise rotation of ice flow. This rotation is most pronounced closer to the former shear margin with the Western Glacier Tongue, where large horizontal gradients in ice flow and the associated lateral drag pushed the ice flow on the Eastern Ice Shelf towards the east. Both the observed and modeled deceleration of ice velocity is further supporting this finding, because reduced lateral drag is slowing down the

Eastern Ice Shelf. This experiment also shows that the Eastern Ice Shelf has not yet fully decoupled from the influence of the remaining parts of the Western Glacier Tongue; measured streamlines in 2019 still show an eastward deflection of ice flow that is not present in the modeled streamlines that completely decouple the Eastern Ice Shelf from any influence of the Western

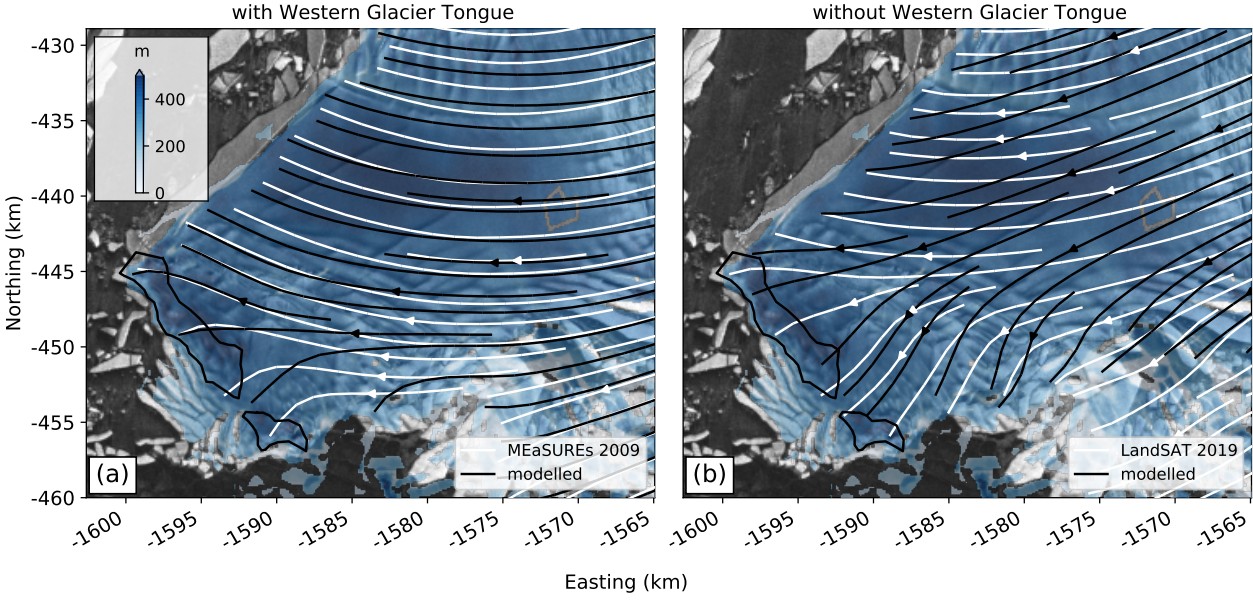

**Figure 8.** Comparison between (white) satellite derived streamlines and (black) numerical solutions from ISSM: (a) The situation before the break-up of the Western Glacier Tongue in 2009 with a relatively well-grounded pinning point. (b) Removing the Western Glacier Tongue from the model domain, and thus weakening its influence on the Eastern Ice Shelf, reproduces the flow divergence around the pinning point as observed in the satellite data, but over-estimates the counter-clockwise rotation of ice flow. This suggests that the Eastern Ice Shelf has not yet fully decoupled from the Western Glacier Tongue, particularly closer to the grounding line. Blue colors correspond to the ice thickness estimate derived in this study.

Glacier Tongue (Fig. 8 b). With continued decoupling of this shear margin in the years to come, it can be expected that the Eastern Ice Shelf will continue to rotate until streamlines align perpendicular to the orientation of the pinning point.

The ISSM experiments further underline the importance of the pinning point on the regional stress balance and thus on the ice-flow dynamics on the Eastern Ice Shelf. Although absolute ice velocities have decreased when compared to those in MEaSUREs product for 2009, we attribute the deceleration to weakening of the shear margin with the adjacent Western Glacier Tongue and not to potential regrounding of relatively thicker ice on the pinning point through advection, because there is no evidence of thicker ice upstream (Figs. 2 and 8). Continued observations, especially of an increase in ice flow velocities across

the bathymetric saddle at the pinning point are necessary to assess the future state of the ice-shelf system, now that the majority of the Western Glacier Tongue's lateral pull on the Eastern Ice Shelf has been removed.

Our model results partially contradict a similar perturbation experiment of Nias et al. (2016), which suggested that removal of the pinning point does not substantially decrease ice-shelf buttressing. We attribute this discrepancy to the coarse resolution of previous bathymetry estimates, which underestimated the elevation of the seafloor ridge underneath the pinning point, as

well as the depth of the trough underneath the Eastern Ice Shelf. The height of this ridge changed from about -800 to -600 m

in the BedMap2 dataset (Fretwell et al., 2013) to -400 to -200 m below the WGS-84 ellipsoid in our adjusted bathymetry. For a given surface elevation, halving the ice thickness beneath sea level and thus reducing hydrostatic uplift doubles the corresponding height above flotation and thus the potential of the net buttressing force exerted on the ice shelf. When an ice surface is well above the buoyancy level, a thicker ice column acts as an obstacle to ice-shelf flow. Furthermore ocean tides are less likely to lift the ice and cause ephemeral grounding. For reference, ocean tides within the Amundsen Sea are diurnal and reach amplitudes up to $\pm 1$ m during fortnightly spring tides (our GPS measurements, Padman et al., 2002). Increased ice-shelf buttressing, in turn, stabilizes the location of the grounding line (e.g., Dupont and Alley, 2005) if no other forcings are changed such as an increase in sub-ice-shelf melting or hydrofracturing.

The shape of the pinning point and its orientation within the flow field is also important to define its role in ice-shelf buttressing. Streamlined ice rises and rumples provide less stabilization than obliquely oriented grounded areas (Still et al., 2019). Although funneling of the ice between the two remaining portions of the pinning point increases lateral shearing, which itself increases the net resistance to ice-shelf flow, shearing also generates rifts, and a preferred crystallographic orientation develops with cumulative strain, such that net resistance decreases compared to initially isotropic poly-crystalline ice (MacAyeal et al., 1998; Hulbe and Fahnestock, 2007; Hudleston, 2015). The magnitude and direction of lateral drag can be calculated using a force budget approach, indicating that effective resistance does not only depend on spatial extent and associated height above flotation of the pinning point but also on the properties of the seafloor materials (Still et al., 2019). With a clear upstream thickening signal (XI in Fig. 5 c) that we attribute to relative compression preceding the redirection of ice flow around it, the smaller western portion of the pinning point is therefore either a sticky spot or frozen to solid bedrock rather than grounded on softer sediments.

Similarly, our inversions of surface velocity fields for the rheological parameter miss the weakening in the small shear zone upstream of the pinning point (Fig. A2) that is evident in visible satellite imagery (Fig. S8). As a consequence, our model simulations are incapable of reproducing the observed velocity field within the limits of the basal friction coefficient. In reality, localized deformation within the narrow shear zone upstream of the pinning point (Alley et al., 2021) is likely the source of mechanically weakened ice that is missed in the model simulations. This increases longitudinal stresses against the pinning point that the model compensates by a four-fold increase of the basal friction coefficient to slow down the ice flow. A more sophisticated model that accounts for ice fracture would help evaluate these factors and distinguish their impact on the flow.

The employed SSA is in line with the analysis of Barnes et al. (2021) for glaciers draining into the Amundsen Sea. While higher-order models are better suited to describe areas of transitional flow, such as ice flow across the grounding line, the SSA is the appropriate approximation to model ice-shelf flow and other areas of low basal friction (MacAyeal, 1989). Similarly, the employed Budd-type friction law (Budd et al., 1984) is only one among several other formulations often used in Antarctica (Joughin et al., 2019). A model inversion using a Weertman-type friction law (Weertman, 1957) would increase the values of the derived basal friction coefficient, because it neglects the effective pressure $N$ at the ice-bed interface. Other possible friction laws subsume the role of $N$ and become essentially plastic at high water pressure, but still yield low values of the basal friction coefficient for low water pressure. That effect is captured in Budd-type, Schoof-type (Schoof, 2005) and other regularized Coulomb laws and we therefore do not expect the choice of friction law to significantly influence our results.

Delineating grounding lines from height above flotation is generally considered less accurate than estimates from double-differential interferometric synthetic aperture radar, because the assumption of hydrostatic equilibrium is violated in the grounding zone (Brunt et al., 2010). With the absence of temporally-dense SAR data acquisition that is required by the fast-flowing Thwaites Glacier, only dedicated satellite missions can provide the necessary coherence to allow the application of InSAR (e.g., Milillo et al., 2019). The highly dynamic character of Thwaites Glacier's response to the ongoing destabilization of the ice sheet-shelf system therefore stresses the use of alternative techniques, such as analysis of height above flotation as a proxy for ice-shelf buttressing and grounding-line retreat (Dupont and Alley, 2005). The launch of Sentinel 1c that is currently scheduled for 2022 will provide the necessary temporal resolution to monitor the anticipated dynamical changes on Thwaites Glacier. Similarly, tidal oscillations of the grounding line were not captured with ICESat-2's repeat pass of 91 days, which is locally significant in areas where bedrock slopes are shallow, such as along the main trunk of the Western Glacier Tongue (Milillo et al., 2019). Complementing InSAR with height-above-flotation-derived grounding lines allowed us to refine previously published retreat rates. An example is the 'butterfly' region where the reported rapid retreat up to 1.2 km yr$^{-1}$ between 2011 to 2017 (Milillo et al., 2019) was narrowed down to have occurred mainly between 2011 to 2014. Furthermore, the calculation of height above flotation is independent of the presence of tidal surface flexure that is required to delineate grounding lines from the analysis of InSAR data. This enabled us to locate two areas landward of the present-day grounding line, which are already at flotation because of depressions in the bed (Fig. 3 b). Continued retreat across the seafloor saddles that currently prevent the intrusion of seawater into these areas will spark an inland grounding-line jump of up to 3 km.

Although our analysis of changes in height above flotation shows the spatial distribution of grounding-line retreat and surface lowering, it does not allow direct conclusions about pathways of warm Circumpolar Deep Water intrusion. New observations collected by an autonomous underwater vehicle reveal that warm water penetrates into the sub-ice-shelf cavity from both sides of the pinning point (Fig. 1 c, Wåhlin et al., 2021). These water masses then converge beneath the Eastern Ice Shelf where a density-driven overturning circulation induces vertical mixing which, in turn, enhances the heat transport from the ocean to the ice base. Cold and freshened meltwater then exits the sub-ice-shelf cavity to the west. This insight highlights the vulnerability of the pinning point to non-linear processes, because the eastern branch of warm water intrusion originates in Pine Island Bay where local meteorological conditions drive variability in ocean forcing of Pine Island Glacier (Webber et al., 2017). Basal melting may be pronounced in the vicinity of the pinning point, because the western branch of warm water intrusion is characterized by southward inflow for all depths across the seafloor ridge (Wåhlin et al., 2021). This hypothesis is supported by ice thinning against the ice-flow direction in the airborne radar transects (Fig. 2), as well as local weakening observed in visible satellite imagery (Fig. S8). Upstream thinning, in turn, reduces the likelihood of regrounding of thicker ice through advection onto the pinning point.

Ocean temperature variability was also linked to the timing of retreat and intermittent re-advance of the Western Glacier Tongue within the last two decades (Miles et al., 2020). A period of extremely warm ocean temperatures coincides with rapid acceleration of the Western Glacier Tongue between 2006 to 2012, followed by cooler ocean temperatures during deceleration between 2012 to 2015. Miles et al. (2020) hypothesize that this intermittent slow down is due to strengthening and re-advance of the shear margin with the Eastern Ice Shelf and is not linked to regrounding of thicker ice on former pinning points. Our

analysis of height above flotation in 2014 supports their hypothesis, because we do not observe re-grounding near the ice edge of the Western Ice Tongue in REMA (Fig. 3 a). Since 2016 the shear margin has weakened, causing rapid acceleration and wide-spread break-up of the Western Glacier Tongue to an ice-mélange that is bound together by landfast sea ice. A large tabular iceberg, named B-22a, which calved off the Western Glacier Tongue in 2002 and is since caught on the seafloor in the Thwaites embayment, might control the persistence of sea ice in this area and thus the integrity of the remainder of the Western Glacier Tongue (Miles et al., 2020). Removal of this iceberg and subsequent loss of landfast sea ice is not only likely to modify regional ocean circulation, but an open-water regime might also allow the seasonal inflow of solar-heated surface water that increases basal melting (Stewart et al., 2019).

Our analysis of height above flotation and surface-lowering rates from ICESat-2 between 2014 to 2020 shows that ungrounding of the pinning point is possible within a decade. Although the Eastern Ice Shelf might retain contact to its pinning point for up to three decades at its maximum height above flotation of >20 m, the present nonlinear processes such as the advection of structurally weak ice onto the pinning point and feedback mechanisms involving oceanic variability in the Amundsen Sea might accelerate this trend. Entire unpinning of the Western Glacier Tongue from a peak on the seafloor ridge between 1996 (Rignot, 2001) and 2011 (Rignot et al., 2016) preceded its break-up from a structurally-intact ice tongue to a mélange of smaller fractured icebergs. We therefore see the break-up of the Western Glacier Tongue as an analog for the time post-unpinning of the Eastern Ice Shelf.

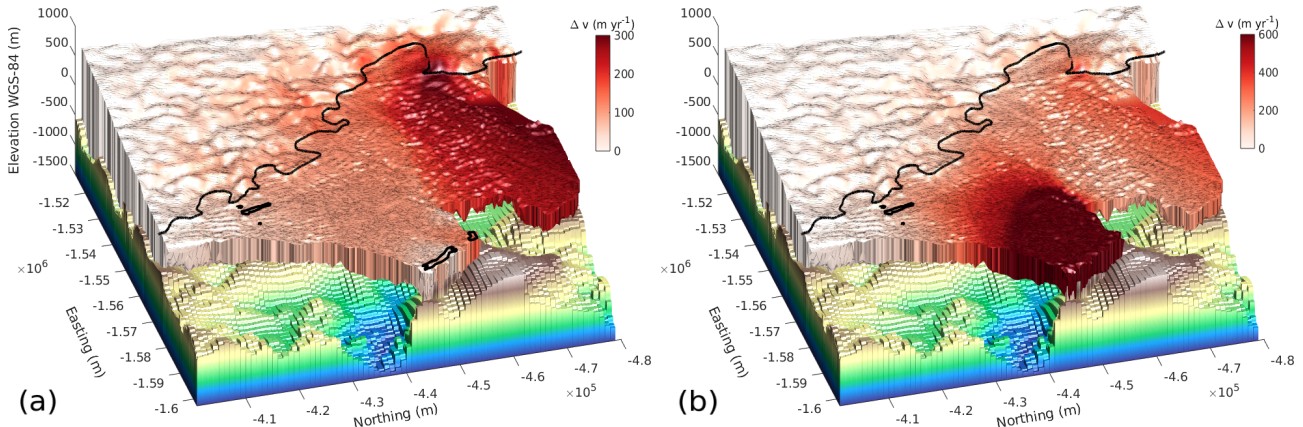

**Figure 9.** The effect of sustained ungrounding between the two remaining portions of Thwaites pinning point on regional ice dynamics, as modeled with ISSM. The black line shows the 2011 grounding line. (a) The black line shows the spatial extent of ungrounding and is guided by our analysis of height above flotation from ICESat-2. Note the funneling of ice over the bathymetric saddle and the increase in ice flow velocities on the Eastern Ice Shelf and further upstream. (b) Modelled acceleration following the entire ungrounding of the Eastern Ice Shelf from the pinning point.

# 5 Conclusions

We present two maps of height above flotation to assess changes in structural stability of the Eastern Ice Shelf between 2014 and 2019/20 by integrating ICESat-2 altimetry data with recent airborne radar surveys. The results show asymmetric grounding-line retreat along the coast accompanied by expansion of newly formed cavities around a series of emerging ice rises sitting on local seafloor highs. The largest retreat rates of up to 1.5 km yr$^{-1}$ are found within rapidly retreating narrow cavities. The pinning point that is currently constraining the Eastern Ice Shelf has separated into a quickly destabilizing eastern portion and a smaller, but structurally stronger western portion as concluded from a pronounced upstream ice thickening signal.

Guided by airborne radar data, our estimate of the height of the seafloor ridge that comprises the pinning point is several hundreds of meters higher than in previously published data sets. This emphasizes the role of the pinning point in buttressing of Thwaites Glacier and will help to improve models used to predict its retreat and sea-level contributions.

Since the break-up of the Western Glacier Tongue in 2009, the ice flow on the Eastern Ice Shelf has slowed down, rotated counter-clockwise, and now funnels through a bathymetric saddle between the two remaining portions of Thwaites pinning point. Model simulations using ISSM reproduce this counter-clockwise rotation of ice flow with the removal of the Western Glacier Tongue from the stress balance and attribute the satellite-observed ice funneling to weakening of the pinning point and opening of the saddle. Whether the consequent flow acceleration is short-lasting or causes sustained ice speedup across the grounding line can not be answered with our diagnostic experiments. Transient model simulations with evolving geometry and ice flow are necessary to shed further light on the future dynamics and stability of Thwaites Glacier. Given current rates of surface lowering from ICESat-2 laser altimetry data in this area, in combination with the advection of thinner and mechanically damaged ice upstream, Thwaites pinning point could reach flotation within less than one decade with implications for the stability of the Eastern Ice Shelf and thus the whole Thwaites Glacier.

Compared to other proposed scenarios of ice-shelf break-up in the near future, such as cracking through the central part of the Eastern Ice Shelf or failure along rifts within the narrow shear zone upstream of the pinning point, our analysis strongly supports that unpinning of the Eastern Ice Shelf from the seafloor ridge and subsequent loss of its structural integrity as a short-range mechanism for break-up. We conclude that unpinning within the next decade, followed by break-up similar to the Western Glacier Tongue, is a very likely scenario of regional destabilization in light of the involved meteorological, glaciological and oceanographic processes. Following other ice shelf disintegration events (e.g., Scambos et al., 2004; Rack and Rott, 2004) we expect increased ice discharge of up to 10 % along a 45-km stretch of the grounding line thereafter.

*Data availability.* Data sources are referenced in the text. Derived grounding-line products are available through the US Antarctic Program Data Center (Wild et al., 2021, https://doi.org/10.15784/601499).

## Appendix A: Parameter estimation and L-curve analysis

We define, $\mathscr{I}(\mathbf{p})$, a cost function that both describes, $I(\mathbf{p})$, the misfit between the model output and the observations and, $R(\mathbf{p})$, the roughness in, $(\mathbf{p})$, the desired model parameter field. The misfit consist of two terms, the absolute and the logarithmic difference between the surface velocities, respectively, where $v_0$ is a minimum velocity to avoid numerical instabilities of

a possible division by zero. The third term in the cost function is either the gradient of the rheological parameter over the entire ice surface ($\mathbf{p} = \mathbf{B}$) or the absolute gradient of the basal friction coefficient underneath the grounded ice ($\mathbf{p} = \beta^2$). The contributions of the three terms to $min_{\mathbf{p} \in \mathbf{P}}(\mathscr{I}(\mathbf{p}))$ are weighted individually with $w_1 = w_2 = 1$, where $w_3$ the Tikhonov regularization parameter, which needs to be tuned to avoid overfitting of $\mathbf{p}$ during the model inversion.

$$\mathscr{I}(\mathbf{p}) = I(\mathbf{p}) + R(\mathbf{p}) \tag{A1}$$

$$= w_1 I_{\mathrm{abs}}(\mathbf{p}) + w_2 I_{\mathrm{log}}(\mathbf{p}) + w_3 R(\mathbf{p}) \tag{A2}$$

$$= w_1 \int_S \frac{1}{2} \left[ \left( \mathbf{v_x}^{\mathrm{mod}} - \mathbf{v_x}^{\mathrm{obs}} \right)^2 + \left( \mathbf{v_y}^{\mathrm{mod}} - \mathbf{v_y}^{\mathrm{obs}} \right)^2 \right] dS + \tag{A3}$$

$$w_2 \int_S \left[ \log \left( \frac{\|\boldsymbol{v}^{\mathrm{mod}}\| + v_0}{\|\boldsymbol{v}^{\mathrm{obs}}\| + v_0} \right) \right]^2 dS + \tag{A4}$$

$$w_3 \int_S \frac{1}{2} \|\nabla \mathbf{p}\|^2 dS. \tag{A5}$$

Generic L-curve analysis is a method to estimate the trade-off between the cost function and the regularization term. It

consists of calculating $\mathscr{I}$ and $R$ for a range of values for $w_3$ displayed on a log figure, where the corner corresponds to the optimal regularization parameter. We follow the iterative procedure by Still and Hulbe (2021) and firstly invert for the rheological parameter with a uniform basal friction coefficient of $\beta^2 = 0$ (Pa yr/m)$^{1/2}$ for floating ice and $\beta^2 = 200$ (Pa yr/m)$^{1/2}$ for grounded ice. We secondly invert for friction using the rheology result from the first step to estimate a spatially variable friction field for grounded ice. Both inversions are performed for a wide range of values for the Tikhonov regularization parameter

from which we choose $w_3^{\mathrm{B}} = 7.9 * 10^{-19}$ and $w_3^{\beta^2} = 3.2 * 10^{-7}$, respectively, for further processing (Fig. A1). We then reset the rheology to a constant value of $\mathbf{B} = 1.8 * 10^8$ (Pa/s)$^{1/n}$ and re-perform its inversion using the improved friction field from the second step. Finally, we reset friction to it's initial uniform field and re-perform its inversion using the improved rheology to also get a refined friction field. We perform this iterative procedure and the L-curve analysis for both the MEaSUREs and Landsat-8 velocity fields and call the respective results 'control run' for the perturbation experiments.

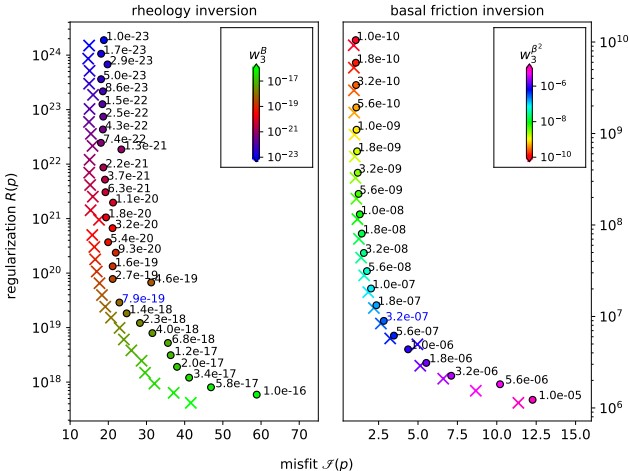

**Figure A1.** L-curve analysis to derive best values for the regularization parameter for the rheological parameter and the basal friction coefficient. The chosen values are displayed in blue. Crosses show results from the MEaSUREs velocity field, circles from our Landsat-8 velocity field.

*Author contributions.* CTW led data analysis, modelling and writing. KEA derived the velocity records from feature tracking and assisted in analysis of ICESat-2 and REMA data. AM guided the bathymetry model adjustment. MT processed GPS data. TAS and ECP assisted in data analysis and design of the study. All authors conducted field work, contributed to writing and approve the final manuscript.

*Competing interests.* The authors declare that the research was conducted in the absence of any commercial or financial relationships that could be construed as a potential conflict of interest.

*Acknowledgements.* We would like to acknowledge the tireless commitment of the rest of the TARSAN on-ice team Bruce Wallin, Douglas Fox, Dale Pomraning and our field guides Cecelia Mortensen and Blair Fyffe. We thank John Paden for assistance with the airborne radar data and acknowledge the use of data products from CReSIS generated with support from the University of Kansas, NASA Operation IceBridge grant NNX16AH54G, NSF grants ACI-1443054, OPP-1739003, and IIS-1838230, Lilly Endowment Incorporated, and Indiana METACyt Initiative. We also thank Laurie Padman for discussions regarding tidal analysis, and Paul Summers for a discussion of friction laws. The authors appreciate the support of the University of Wisconsin-Madison Automatic Weather Station Program for the data set, data display, and information (NSF:Grant 1924730). This work is from the TARSAN project, a component of the International Thwaites Glacier Collaboration (ITGC). Support from National Science Foundation (NSF: Grant 1929991) and Natural Environment Research Council (NERC: Grant NE/S006419/1). Logistics provided by NSF-U.S. Antarctic Program and NERC-British Antarctic Survey. ITGC Contribution No. ITGC-045. CTW also thanks Wolfgang Rack for providing access to the COMSOL Multiphysics finite-element software. We thank Josefin Ahlkrona

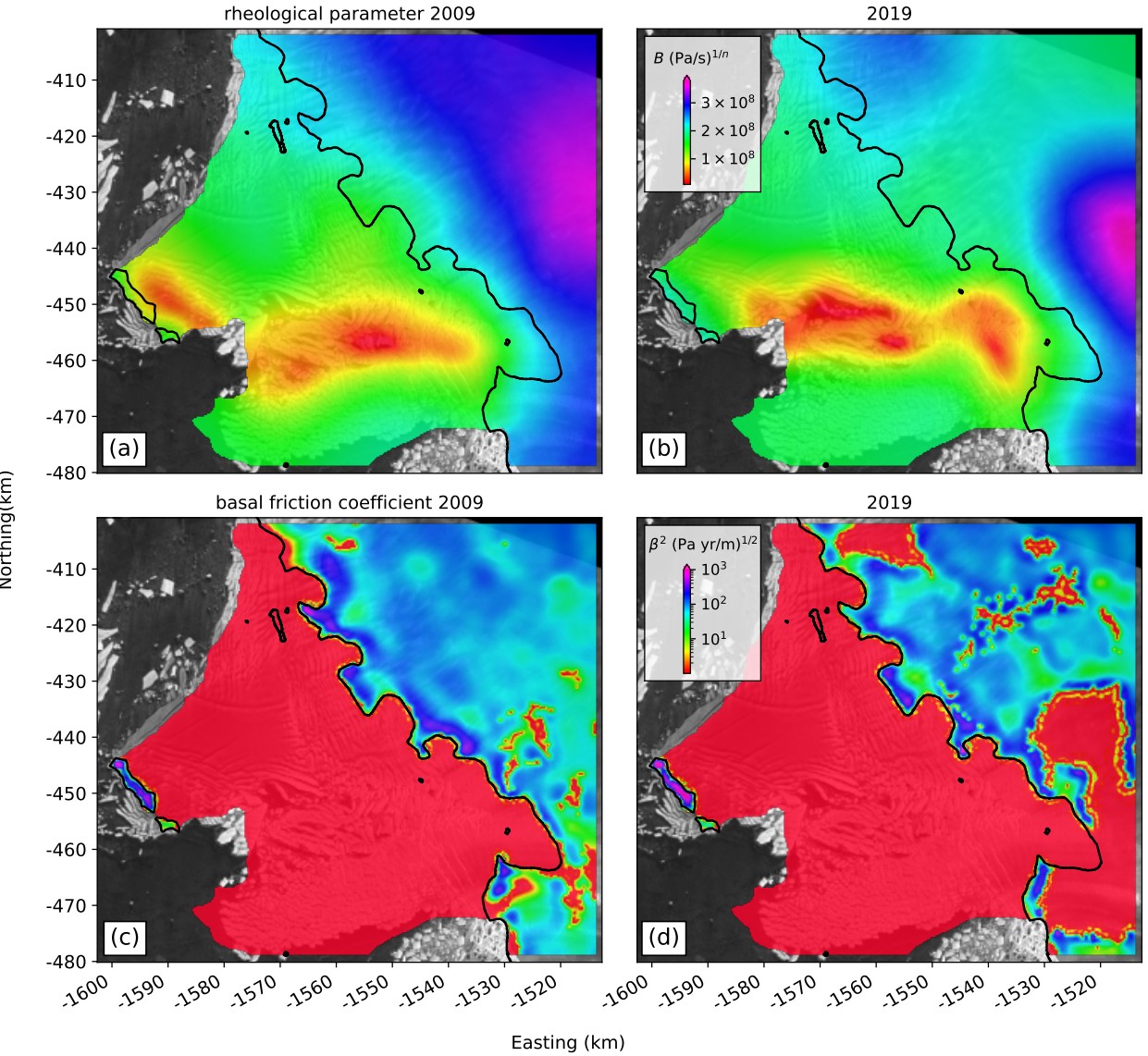

**Figure A2.** Inferred fields of the rheological parameter and the basal friction coefficient from the iterative inversion using both the MEa-SUREs and our LandSAT-8 velocity field. Note: (i) the rheological weakening of the shear margin between the Eastern Ice Shelf and the Western Glacier Tongue, (ii) the misrepresented rheological strengthening of the shear zone upstream of the pinning point that is clearly absent in visible satellite imagery, (iii) high values of the basal friction coefficient at the pinning point and (iv) very low friction values underneath Thwaites Glacier's main trunk that suggest distributed thawed bed properties.

for editing, Clemens Schannwell and a second anonymous referee for their constructive comments that significantly improved the quality of this manuscript.

**Table A1.** Symbols used in this study

| Tidal-flexure modeling | | |
|---|---|---|
| **w** | elastic vertical deflection | |
| $k$ | spring constant of bed | 5 MPa m$^{-1}$ |
| **D** | ice-shelf stiffness | |
| **q** | tidal force | |
| **H** | ice thickness | |
| $E$ | Young's modulus | $1.5 \pm 0.69$ GPa |
| $\lambda$ | Poisson's ratio | 0.4 |
| $A$ | tidal amplitude | 1 m |
| $g$ | gravitational acceleration constant | 9.81 m s$^{-2}$ |
| **Height above flotation calculation** | | |
| $\mathbf{z_f}$ | freeboard | |
| $\mathbf{H_f}$ | ice thickness in hydrostatic equilibrium | |
| $\mathbf{H_a}$ | absolute ice thickness | |
| $\rho_{sw}$ | ocean water density | 1027 kg m$^{-3}$ |
| $\rho_{ice}$ | ice density | 917 kg m$^{-3}$ |
| **Ice-dynamics modeling** | | |
| $\boldsymbol{\sigma}_{ij}$ | deviatoric stress tensor | |
| **B** | rheological parameter | |
| $n$ | creep exponent | 3 |
| $\dot{\epsilon}_e$ | effective strain rate | |
| $\dot{\boldsymbol{\epsilon}}_{ij}$ | strain rate tensor | |
| $\boldsymbol{\tau_b}$ | basal drag force | |
| $\beta^2$ | basal friction coefficient | |
| **N** | effective water pressure | |
| $\boldsymbol{v_b}$ | basal velocity vector | |
| $\mathbf{z_b}$ | bathymetry | |
| $p_{sw}$ | sea water pressure | |

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
