# Peer review of "Weakening of the pinning point buttressing Thwaites Glacier, West Antarctica"

_The Cryosphere, 2021_

## Referee Comment (RC1)

**Review of Wild et al. "Weakening of the pinning point buttressing Thwaites Glacier, West Antarctica"**

June 3, 2021

**General comments:**

The manuscript by Wild et al. presents a very detailed evaluation of a pinning point that is currently buttressing Thwaites Glacier. Using a range of different tools (e.g. remote sensing data, diagnostic ice-sheet modelling, etc.), the authors reconstruct the evolution of the pinning point and its effect on the ice shelf and its overall integrity. They conclude that it is likely that the ice shelf will detach from the pinning point within the next decade. They hypothesise that detachment of the pinning point will lead to acceleration upstream of the grounding line and could lead to the disintegration of the Western Glacier Tongue.

Given the importance of the Amundsen Sea Sector for sea-level rise over the coming decades and centuries, I find the topic highly relevant to the cryospheric community. The paper is well written and Figures are appropriate. I also want to commend the authors on utilising different complementing methods to provide such a detailed analysis. This is certainly a strong point of the study. The study also highlights the importance of pinning points either as ice rises or rumples.

I support publication in TC. I have, however, a few general comments concerning the ice-sheet model inversion, and a number of detailed comments that should ideally be addressed. I hope the authors find my comments helpful.

**Specific comments:**

**Main concerns:**

1. In my view, the presentation of the numerical inversion part of the study is relatively weak in its present form, primarily because many details are omitted. This section certainly needs more explanation and I must admit that I am not really convinced of the basal traction field that the authors provide in the supplement (Fig. S5). I

am happy with the fact that the authors describe the ice-sheet model in brief without going into to much detail about the underlying equations, but nevertheless the basic setup including parameters and boundary conditions should be provided. I find the description of the inversion and model insufficient.

- You are saying that you are using SSA to invert for basal friction and then solve BP and FS in a diagnostic run (L214-216). Do you actually use different stress models in your diagnostic runs? This is never mentioned again. If you do use different stress models for inversion and diagnostic run (e.g. SSA for inversion and FS for perturbation), why should this be justified? I found in my work (admittedly in forward simulations) that using SSA for the inversion and FS for a forward simulation certainly results in a larger model drift. The reason being that you are simply not inverting for the same stress model.

- I am a bit concerned that you overfitted your data during the inversion. Your Fig. S5 looks very patchy, so from my experience this is what basal traction inversion fields look like after a few iterations (not converged yet) or if you do not regularise enough (or at all?). In any case, you need to add what kind of regularisation you use and also add an L-curve analysis to convince me that you are not overfitting (see Fürst et al. 2015 or Berger et al. (2016) as an example). A 2D plot of the mismatch between modelled and observed velocities would also be appreciated. This can be added to the Supplementary material.

- I am also wondering why you are just inverting for the basal friction coefficient, but not for ice viscosity at the same time which is actually pretty standard these days when using SSA (see for example Fürst et al. 2015, Cornford et al. 2015). This would also help to identify regions of shelf weakening in response to your pinning point being there (see Berger et al. (2016)) and also how these regions have changed as the size of the pinning point has changed.

- Please also add what kind of ice temperature you employ? Is your setup isothermal? What is your rate factor? All these things are needed for a complete description of you setup.

**Technical corrections:**

Abstract:
L5 I would change "model perturbation" to "ice-sheet model perturbations" to be more precise

L43 ...emergent ice rise ... This could use a citation

L72 Similar to L5, I would suggest to change it to "ice-sheet model perturbation experiments"

L86 How come the Jordan et al. (2020) map has resolution of 1000 m, but you say in L82 that airborne gravimetry data cannot resolve anything below 5000 m?

L97-105 How do you ensure that you have a smooth transition (no jumps) between the region where you adjust the bathymetry and the background bathymetry?

L130 Equations: Here you write explicitly that $\mathbf{w}$ is a function of space. Either you do this everywhere (Eq. 4 and 5) or you drop it in all equations. Just be consistent. I think I would prefer the former but leave this to the authors to decide.

L135 I think it would really help to add a table with symbols/constants and values used in the study. Coming from ice dynamical modelling, the letter $\mathbf{q}$ is reserved for ice flux, so I think adding a table with all symbols and constants would be beneficial. You could then also drop the values in the main text.

L155 Is there also an uncertainty associated with REMA?

L178 How do you get the uncertainty in grounding-line retreat rates? Does that not depend also on the geometry of the region (e.g. less inclined bed being more uncertain?)?

L191 Again, can you briefly say how you arrive at these numbers?

L213 Shouldn't that be "Shelfy"?

L223 How can your mesh be 100 m resolution, if you are using mesh adaptation? I think you should move this to the end of the previous paragraph (L218) and give a max/min resolution range.

L224-225 I would delete the sentence starting with "We set grounded ...". You could set it to 1000 m/yr, but if you are running diagnostic experiments without geometry updates, this does not matter.

L230 I think here or somewhere nearby, it would be good if you could add what kind of boundary conditions you are applying in your experiments.

L254 How do you compute grounding-line retreat rates of ice rises? Is this a maximum estimate or a mean estimate? Would be good to add this.

L311 If I understand the reported data here correctly (8.0 HAF and 0.3 m/yr thinning), I find your formulation in the abstract and conclusions a bit too certain ("will unpin in less than a decade"). I think based on the extrapolation of your data it's more likely

to happen within the next 20-30 years. So I would suggest to add a qualifier to these statements along the the lines "with the potential to detach within the next decade"

L313-315 To interpret this we certainly need for information on regularisation etc. See main concerns.

L320 What is "the isotropic consideration"? Do you mean that your rheology assumes that ice is isotropic?

L324 You need to say somewhere what stress model you use for you diagnostic simulations, but ideally before in section 2.5

L325 Here you introduce the acronym ISSM, but you have used this already before. Please make sure that you introduce acronyms at first mention. I am not sure exactly what the TC policy is, but REMA and I believe MEaSUREs were never introduced. I am familiar with the data products, but I am not sure if journal policies require you to introduce these as well.

L336-337 Based on the evidence that you show, we know that the direction is correct. What about the magnitude? I think adding a 2D velocity magnitude difference plot would be helpful.

L337 When you remove the Western Glacier Tongue, do your boundary conditions change? What boundary condition do you apply where the Glacier Tongue was removed?

L348 Are you still using the same basal friction field or did you rerun the optimisation with the updated velocity dataset? If the former, please add a point of why you think this is justified?

L375 What are "fast horizontal gradients?" Do you mean large? I do not think gradients can be fast.

L377 I just do not quite follow, how can reduced lateral drag result in slower ice velocities? Could you please explain this a bit more?

L387 Why can it not be thicker ice that is being advected? Is there no evidence of thicker ice upstream?

L401 I would change to ...if no other forcings are changed such as ..."

L416-417 The reported limits of the basal friction coefficient? I do not understand this sentence. Also as mentioned previously, you could incorporate anisotropic effects for the ice shelf into the model the same way you do for the basal friction coefficient, if you also invert for ice viscosity.

L432 I agree that it is always good to have multiple complementing methods. However, I feel that it should be mentioned somewhere that your flotation assumption is violated at the grounding line and hence will never give you the "true" grounding line location. Also I found that height above flotation is also quite sensitive to density estimates of the ice column and ocean.

L476 delete vigorous

L488 I would add a qualifier here: "could reach flotation ..."

**Figures:**

Fig. 2: Could you indicate ice flow direction also in panel (a). The dashed white line is also not visible in the legend

Fig. 3: How did you compute retreat rates? Along flowlines? Maximum retreat? Especially retreat for region IV in panel (c) looks strange. I am also not sure if panel (a) is needed. You could potentially delete this and change the layout to Fig. 5

Fig. 6: The letters U and D are hard to read. Can you change them to black font colour?

Fig. 7: I like these 3D Figures. But can you add either a flow direction arrow or indicate in the caption that 3D Figures are rotated by 90 degrees? Or if this is possible, have the same orientation as all your 2D maps?

Fig. 9: I can hardly see the red region. Since your colour scale includes red, please change to something different.

Sincerely,
Clemens Schannwell

**References**

Berger, S., Favier, L., Drews, R., Derwael, J., and Pattyn, F. (2016). The control of an uncharted pinning point on the flow of an Antarctic ice shelf. Journal of Glaciology, 62(231), 37-45. doi:10.1017/jog.2016.7

Cornford, S. L., Martin, D. F., Payne, A. J., Ng, E. G., Le Brocq, A. M., Gladstone, R. M., Edwards, T. L., Shannon, S. R., Agosta, C., van den Broeke, M. R., Hellmer, H. H., Krinner, G., Ligtenberg, S. R. M., Timmermann, R., and Vaughan, D. G. (2015). Century-scale simulations of the response of the West Antarctic Ice Sheet to a warming climate, The Cryosphere, 9, 1579–1600, https://doi.org/10.5194/tc-9-1579-2015

Fürst, J. J., Durand, G., Gillet-Chaulet, F., Merino, N., Tavard, L., Mouginot, J., Gourmelen, N., and Gagliardini, O. (2015). Assimilation of Antarctic velocity observations provides evidence for uncharted pinning points, The Cryosphere, 9, 1427–1443, https://doi.org/10.5194/tc-9-1427-2015

---

## Referee Comment (RC2)

**Review of tc-2021-130**

**General comments**

Wild et al. study recent changes in ice dynamics of Thwaites Glacier in West Antarctica and investigates the impact of a subglacial pinning point under the eastern part of the Thwaites Ice Shelf. The authors use a suite of satellite imagery and ice-penetrating radar data to characterize changes over the last decade. A numerical ice sheet model is used to perform diagnostic (fixed-geometry) experiments to understand the ice dynamics associated with recent changes, and to assess the consequences of potential near-future ungrounding of the pinning point.

Thwaites Glacier is a major contributor to sea-level rise in West Antarctica, a region where there is potential for non-linear future mass loss. The underlying processes are not well constrained and the study is thus a timely contribution to a growing body of research on the glaciers in the Amundsen Sea, West Antarctica. The study clearly fits within the scope of TC and should be of high interest for the glaciological and broader scientific community.

The manuscript is well written with an overall coherent story. In general, the datasets and observational methods used are thoroughly described and the conclusions are supported by the analysed data and model experiments. Overall I think this is an exciting study with some unique datasets and interesting model experiments giving us an improved understanding of ice-shelf-glacier behavior at Thwaites.

The few major comments I have relate to the numerical model setup and description, as well as the need for a broader perspective on previous literature in the Introduction, and considering wider implications in Discussion. These comments should be addressed by the authors before publication in TC can be considered.

I should mention that my expertise is mainly ice dynamics and ice-sheet modelling, rather than remote sensing and radar glaciology. My comments below reflect this; the Editor should perhaps consider perspectives from a more empirical-minded scientist as well to get a balanced overall evaluation of the paper.

**Major comments**

*Model setup and description*
The following comments largely regards Section 2.5 Ice-dynamics modelling. While this section already contains useful information about the model, it is in my view not

yet complete and does not allow the reader to fully understand your model setup and experiments. As it stands, this does somewhat erode the strength of your findings about the importance of the pinning points and the potential effects of future ungrounding. To alleviate this, I list below some concrete suggestions which I hope that the authors will consider.

First of all, it would be nice with a brief starting paragraph describing what the rationale is behind using a numerical ice sheet model. As it is written now, this section starts a bit abruptly with the reader not really knowing where we are going.

- Regarding the friction law (Eq. 5), you should mention that you are assuming a linear Budd-type law, and that this is just one among several possible friction laws in the literature, and whether/how you think using other friction laws often used in Antarctica (e.g. a Schoof or Tsai law) would influence your results, if at all (this could also be done in Discussion).
- Need to mention what the assumption is for the effective pressure (perfect hydrological connection between the ocean and the grounded ice base), what the equation for N is (can be done in-text), and also what processes/factors this formulation of the effective pressure neglect.
- It is unclear to me why you use one ice-flow approximation for the inversion of the friction coefficient beta (2D SSA), and another for the stress balance calculation (3D Higher Order). Isn't the inversion for beta essentially a series of stress balance calculations? In my view, the cleanest and most consistent approach would be to use Blatter/Pattyn (Higher Order) for everything. I assume this wouldn't be too computationally demanding since you're only doing diagnostic experiments. If the authors have a very good reason for not using the same ice-flow approximation across all experiments, this needs to be stated clearly. And if you're going from 2D SSA to 3D Higher Order, how do you deal with the boundary conditions? Are you rerunning in 3D once you have your 2D inversion (and why?).
- The description of the model mesh needs to be more specific. You mention that you're using an anisotropic mesh refined near grounding lines and narrow shear zones. What exactly is the mesh resolution here, and what is the resolution elsewhere? If you're using a 3D Blatter/Pattyn approximation, how many vertical layers do you use? Are they uniformly vertically spaced, or do you have more finely spaced vertical layers towards the bed? See also Minor comments below.
- How is grounded ice rheology chosen/inferred? What ice temperature is assumed, based on what? How well constrained is this? What are associated uncertainties and potential influence on results?

- How is ice-shelf rheology chosen/inferred? What ice temperature is assumed, based on what? How well constrained is this? What are associated uncertainties and potential influence on results?
- A reference to Table 1 in this section is missing; it would be great if you could briefly describe the experimental setup already here. As it stands, you have mostly outlined the model setup, while the details of your different experiments are missing in this section and are instead described in the Results (Section 3.4). One option is to move parts of Section 3.4 that concerns the experimental setup to Section 2.5.

*Description and discussion of model results*
The following comments regard mainly Section 3.4 Numerical modeling of the ice-shelf response. In general this section is interesting and the experiments are potentially very revealing. However I'm missing key information regarding the model (see above) and the experimental setup. Therefore it is difficult to fully evaluate the insight gained by this modeling exercise.

You artificially increase basal drag to account for lowered ice viscosity in shear zones; is this straightforward or are there some pitfalls in doing this? Would it be possible to do two separate inversions, inverting for ice rheology (stiffness) for floating-ice areas, and for basal drag for grounded-ice areas? I assume this wouldn't be too much extra work, and wouldn't add much computation time either since you're only doing time-independent stress balance simulations. Adding variable ice-shelf rheology could move your model setup in a more realistic direction and potentially strengthen your interpretation of the importance of the shear margins on ice-shelf flow, and perhaps even reduce the need to increase the friction coefficient 'artificially' so much at the pinning point.

A related question about the inversion and velocity calculations. How well do the modelled velocities actually fit with observed ones, for the 2009 and 2019 observations? What are the RMSEs? Do they consistently under/overestimate? Are there some problematic areas? What about grounded vs floating regions? I think this should be included in a supplementary figure, to give the reader an idea of the model behavior. In particular, we're very interested how well the model captures your key regions of interest (the offshore pinning point, the grounding line, the shear zones). On L336-337 for example you state that "… the inversion is successful…" - more specifics would strengthen this section and further indicate whether your inferences about current and future ice dynamics are robust.

I would like to applaud the authors for the nicely constructed experiments to test the influence of the breakup of the Western Glacier Tongue (Figure 8 and S4).

The 'ungrounding experiments' (Fig 9) are very interesting, but I would be a bit cautious, since your experiments are stress balance (diagnostic) simulations, and we do not know whether this increase in ice velocity will quickly dissipate or whether it will cause sustained ice speedup. Of course if the latter is the case, this will have major implications for the future of Thwaites over the next decades. You say in the Discussion that you see the breakup of the Western Glacier Tongue as an analog for future post-ungrounding of the Eastern Ice Shelf, and you clearly state in the text that you're indeed only doing diagnostic experiments. For balance, I think you should also state (in the abstract and/or conclusions) that model experiments with evolving geometry and ice flow would shed further light on the future dynamics and stability of Thwaites. To be clear, I think that transient experiments are out of scope of the current paper, and I do not expect the authors to include this in a revised version of the manuscript.

*Wider implications*
I would like to see some more elaboration on previous work/state-of-the-art in the Introduction, and related to this, a broader discussion of the study's findings in the Discussion.

The Introduction overall reads well, with a good introduction to pinning points, ice rises and rumples. There is an appropriate background about Thwaites, including both changes over the long-term (Holocene) and the recent (last decades) past. However, as it stands, the Introduction is a little bit too Thwaites-heavy for my taste. There is scope for expansion about other glaciers than Thwaites, and why studying pinning points/ice shelf retreat is vital. There is relevant information about this already in the first two paragraphs, but the rest of the Introduction talks mostly about Thwaites. Of course Thwaites is a key glacier which the community strives to understand, but there are also others (in Antarctica, Greenland, and beyond). What are the particular knowledge gaps in our current understanding of pinning points/ice-shelf buttressing? What have others done before, and what does this study bring that is novel and different? I think by all means the data and findings in the study are novel, it's just that you can set the stage more clearly already in the Introduction, and then later pick up on this in the Discussion.

**Minor and technical comments**

In general, more cross-references to figures and other sections would be great to help the reader.

*Abstract*
L5. "model perturbation experiments" – could be more specific of what kind of model (ice-shelf-glacier model, numerical model, etc)

*1 Introduction*
L51. retreat rates -> please specify if you talk about retreat rates of the grounding line or the ice-shelf front here (I think you mean the former)

*2 Data and Methods*
L86-87. Not entirely clear on the use of bed/bathymetry data here. Please clarify which dataset you use where; how are they integrated/combined? (Jordan et al., 2020 and Morlighem et al., 2020).

L92-93. Would be nice to add a cross-reference to Fig 1a here

L95-96. This is useful information – please specify why (mechanisms/processes) a crevassed (smooth) ice base indicates that ice is in disequilibrium (equilibrium) with the underlying ocean

L168. Which observation? Please clarify.

L182-183. Typo; should be "Lagrangian"

L199. Could you be specific on how large this contribution from REMA is, compared to the others mentioned on L196-197?

L203. should mention what the acronym ISSM stands for, as well as add reference (Larour et al., 2012?)

L213. Typo: should be "Shelfy-"

L218-219. "… guided by surface velocity fields…" – this is a bit vague, please be more specific how you construct the model mesh. Do you only use the gradient of the

observed surface velocity fields (which one), or also set the mesh resolution manually in some areas? You mention on L223 that the horizontal resolution of the final mesh is 100 m; does this mean that your mesh resolution is uniform over the entire domain?

*3 Results*

L232. Figure 3d is not discussed explicitly in this section; perhaps it should be at least cross-referenced to point the reader to this useful summary of temporal evolution of observed retreat rates.

L240-263. What would improve the dynamical insight of this section further is to link these observed changes in grounding-line locations to changes in surface elevation or ice thickness (for example those shown in Figure 5c, but also elsewhere on the larger ice-shelf and upstream of the grounding line).

L265-292. Overall nicely written and interesting findings. Could mention and add cross-references to modeling results (section 3.4) to guide the reader further through this section.

L305. Please explain what a sastrugi is for readers not familiar with snow dynamics.

L308-311. Your final point about ungrounding of the pinning point is important as it is one of the main conclusions of the paper, and even occurs in the title. From the numbers you have provided (avg. height above flotation 8.0 +/- 6.5m and avg. surface-lowering rates 2014-2019/2020 of -0.3 +/- 1.5 m/a), I think you should be a bit more careful with stating that the pinning point may unground in less than one decade. Yes this is possible if the true height above flotation is on the lower end of your stated 8.0 +/- 6.5m, and/or if the true surface-lowering rates are on the higher end of your stated -0.3 +/- 1.5 m/a. Unless I misunderstood something, if you take the average values 8.0 m and -0.3 m/a, complete ungrounding will take 26.7 years. This is of course still interesting and potentially somewhat alarming, but not as dramatic as "less than a decade". If my understanding of what you have done is correct, I think you can phrase this in a more balanced way (as well as in the abstract and conclusions); you should indeed mention what "could" happen, while explaining what is the "best-guess" (average?). I think you can also state more explicitly here (as well as in the abstract and/or in the conclusions) that your inferred timing of ungrounding is merely an extrapolation of recent thinning rates into the future, and not a prediction. Running a numerical model forward in time could obviously shed light on this in more detail.

L313-315. what is the minimum friction coefficient allowed in the inversion, is it 10 (Pa yr/m)1/2 ?

L334. When inferring basal friction coefficients, why do you use the 2011 grounding line for both the MEaSUREs 2009 (Figure S5a) and Landsat-8 2019 (Figure S5b) velocity fields? Why not use the updated grounding lines from 2019/2020? You state on L334 that all simulations use the 2011 grounding line. Please explain what the reasoning behind this is.

L358. I think you mean "model elements" rather than "pixels" ?

*4 Discussion*
Overall very well written, discusses main sources of uncertainty and suggests next steps for improvement, as well as discussing results in light of previous studies. A wider-scope discussion related to other Antarctic glaciers and implications for our understanding of ice-shelf buttressing and ice-shelf dynamics in general would improve this section even further (related to my comments about the Introduction), see also Major comments above.

*5 Conclusion*
L485. I think you should state what kind of model, and that simulations are fixed-geometry velocity snapshots, and thus does not establish whether this will be a short-lived or sustained feature

L493. "… possibly the most likely" – this seems a bit vague or even contradictory. And based on what exactly? You could be more specific here.

**Figures**

In general, figures are nicely crafted and clear.

*Figure 1*
Missing information on what A and B means in (a) – I think they are airborne radar flight lines? Also, in (a) and (c), the colorbar labels should be m above sea level (m a.s.l.)

*Figure 2*
The dashed white line for hydrostatic equilibrium is not visible in the legend

*Figure 7.*
Would be useful if these figures had the same orientation as Figures 1-6 and 8. This also applies for Figure 9.

*Figure S1.* Missing axis labels and units in all panels.

*Figure S4a.* In the legend, "ice (m)" is a bit cryptic. I think you mean "ice thickness" ? In the supplementary, I would also like to see plots of the modelled velocity magnitudes and their mismatches with observed velocity fields (2009 and 2019), not only the velocity directions (see comments above).

---

## Author Comment (AC1)

**Opening statement by the authors, written by Christian Wild (14 Sept'21)**

We thank both reviewers for their respective suggestions on improving the manuscript, and in particular its final modeling part. Our static ice-sheet model experiments are certainly only a first step towards predicting the fate of Thwaites Glacier. Our goal was to identify the main causes behind the observed dynamic changes that are highlighted in the remote-sensing and fieldwork sections earlier in the paper. The motivation for this is that discussions at the annual science meeting of the ITGC 2020 showed that many wonder about the importance of buttressing from the Eastern Ice Shelf on the grounded parts of Thwaites Glacier. As our paper quantifies changes in the pinning point that provides most of the Eastern Ice Shelf's buttressing, we feel it is important to include simple modeling that shows evidence that ungrounding of the pinning point directly affects the stability of Thwaites Glacier even beyond the Eastern Ice Shelf. We would like to leave the question of the temporal evolution after the unpinning to the two dedicated modeling projects within the ITGC (named DOMINOS and PROPHET) that employ state-of-the-art numerical modeling of the future of Thwaites Glacier to quantify changes in ice discharge across a retreating grounding line.

Overall, we have identified three major comments that overlapped between both reviewers: (1) Why invert one ice-flow approximation for the basal friction field and then use another stress balance for forward modeling? (2) Can we invert for basal friction field as well as for ice rheology? (3) Was regularization applied? To address these comments, we have now implemented the model using the SSA for both the inversion and the forward modelling. We present the results of an iterative procedure by Still and Hulbe, 2021 (TC) with an L-curve analysis for the ice rheological parameter B in a first inverse step. We then perform a second inverse step to derive an improved basal friction field with a second L-curve analysis and lastly use the derived fields in a second round of inversions to further improve the estimate (see detailed description below).

The review process has led to the inversion of ice rheology in addition to the inversion of a basal friction field, a thorough L-curve analysis in the Appendix and a set of new figures. We now have two improved control runs for our perturbation experiments (removal of the Glacier Tongue and weakening/ungrounding of the pinning point). This last part of the analysis is still in process and will be included in a revised manuscript. While these improvements have not altered our conclusions, they allow us to present our results with higher confidence. We hope that our efforts were what the reviewers envisioned.

[Figure]

*Illustration 1: Inferred basal friction and ice rheology fields after sequential L-curve analysis and the misfit between modelled and observed velocity. Low values of the rheology show structurally weak ice, such as in the shear margin and the shear zone. This is a control run for further perturbation experiments*

[Figure]

*Illustration 2: Results of the ungrounding experiments using the improved basal friction and rheology fields. The conclusions didn't change and flow acceleration always occurs when parts of/the entire pinning point is ungrounded*

The manuscript by Wild et al. presents a very detailed evaluation of a pinning point that is currently buttressing Thwaites Glacier. Using a range of different tools (e.g. remote sensing data, diagnostic ice-sheet modelling, etc.), the authors reconstruct the evolution of the pinning point and its effect on the ice shelf and its overall integrity. They conclude that it is likely that the ice shelf will detach from the pinning point within the next decade. They hypothesise that detachment of the pinning point will lead to acceleration upstream of the grounding line and could lead to the disintegration of the Western Glacier Tongue.

The reviewer is a specialist in modeling pinning points and has clearly understood the paper. The conclusions are well summarized, but just to be clear, the unpinning will lead to the disintegration of the Eastern Ice Shelf and not the Western Glacier Tongue – but this was probably just a typo anyway.

Given the importance of the Amundsen Sea Sector for sea-level rise over the coming decades and centuries, I find the topic highly relevant to the cryospheric community. The paper is well written and Figures are appropriate. I also want to commend the authors on utilising different complementing methods to provide such a detailed analysis. This is certainly a strong point of the study. The study also highlights the importance of pinning points either as ice rises or rumples.

The reviewer is also aware of the discussions about the importance of Thwaites Glacier in the near future that are at the heart of an ongoing research across the cryospheric community. We thank the reviewer for his very positive feedback regarding the suite of our methods and look forward to possible collaboration in other parts of Antarctica or Greenland, where the tools that are described in this paper might also be useful.

I support publication in TC. I have, however, a few general comments concerning the ice-sheet model inversion, and a number of detailed comments that should ideally be addressed. I hope the authors find my comments helpful.

Yes, your comments were VERY helpful and have greatly elevated the quality of the manuscript and our expertise in ice-sheet modeling.

**RV1-1 Specific comments:**

**Main concerns:**

In my view, the presentation of the numerical inversion part of the study is relatively weak in its present form, primarily because many details are omitted. This section certainly needs more explanation and I must admit that I am not really convinced of the basal traction field that the authors provide in the supplement (Fig. S5). I am happy with the fact that the authors describe the ice-sheet model in brief without going into to much detail about the underlying equations, but nevertheless the basic setup including parameters and boundary conditions should be provided. I find the description of the inversion and model insufficient.

We agree with the reviewer and have now performed two consecutive L-curve analyses for the rheology and the basal friction field. Simultaneous inversion is at least to our knowledge not possible in ISSM. Consecutive inversion of ISSM has shown to produce very similar results as simultaneous inversion of the STREAMICE and Ua models, especially for the basal friction field and rheology on floating parts (Barnes et al., 2020, TC). We now follow the iterative procedure of Still and Hulbe, 2021 (TC) and firstly invert for rheology with a uniform friction coefficient of 200 for grounded ice and 0 for floating ice. We then invert for friction using the rheology result from the first step to estimate a spatially variable friction field for grounded ice. We then reset the rheology to a constant value and re-perform its inversion using the improved friction field from the second step. Finally, we reset friction to it's initial uniform field and re-perform its inversion using the improved rheology to also get an improved friction field. We perform this procedure for both velocity fields and call the results 'control run' for further experiments. The results of the MeaSures control run are shown in illustration 1. Although we can now see weakened rheology in the

sheart margin/zone (indicated by small values of the rheological parameter B), the inferred friction values on the pinning point are still high (<1000)

**RV1-1a** You are saying that you are using SSA to invert for basal friction and then solve BP and FS in a diagnostic run (L214-216). Do you actually use different stress models in your diagnostic runs? This is never mentioned again. If you do use different stress models for inversion and diagnostic run (e.g. SSA for inversion and FS for perturbation), why should this be justified? I found in my work (admittedly in forward simulations) that using SSA for the inversion and FS for a forward simulation certainly results in a larger model drift. The reason being that you are simply not inverting for the same stress model.

We were initially using SSA to invert for basal friction, and then BP in a forward run (never FS). The reason was simply to shorten the amount of time that was required for the inverse step. We have compared the SSA and BP forward solutions and confirm your comment, that the model drifts because of solving a different stress model:

[Figure]

*Illustration 3: Comparison of forward simulations using SSA and BP confirming the suggested model drift by the reviewers. We therefore now use SSA for both the inversion of rheology and friction, and the forward experiments with removing the Glacier Tongue from the domain and ungrounding of the pinning point*

Thanks for raising this concern - we are now following the analysis of Barnes et al. (2020) and use the SSA for both inverse and forward step. However, we interpolate the forward solutions onto a 3D mesh from the original BP analysis only to visualize some of these results in 3D.

**RV1-1b** I am a bit concerned that you overfitted your data during the inversion. Your Fig. S5 looks very patchy, so from my experience this is what basal traction inversion fields look like after a few iterations (not converged yet) or if you do not regularise enough (or at all?). In any case, you need to add what kind of regularisation you use and also add an L-curve analysis to convince me that you are not overfitting (see Fürst et al. 2015 or Berger et al. (2016) as an example). A 2D plot of the mismatch between modelled and observed velocities would also be appreciated. This can be added to the Supplementary material.

Another great comment ! We thank you for this suggestion and pointing us to the relevant literature. Although we did apply regularization in the initial model runs, the L-curve analysis showed that the chosen regularization parameter for basal friction was too small (8e-15 compared to 3.2e-7). Our strict convergence criterion then forced the model to consecutively overfit the data, resulting in a patchy/wiggly basal traction field although the convergence criterion was fullfilled (minimal variation of the cost function between consecutive iterations). The much larger regularization parameter has caused a much more realistic basal traction field with keeping a similar misfit between modelled and observed surface velocities. We now perform two L-curve analysis for rheology and the friction field in the first round of inversions as described above. We've added the figure as requested and a table about the mismatch for different thesholds in velocity.

[Figure]

*Illustration 4: L-curve analysis to derive best values for the regularization parameter for ice rheology and basal friction. The chosen values are colocoded in blue. Crosses show results from the Measures velocity field, circles from our Landsat velocity field.*

**RV1-1c** I am also wondering why you are just inverting for the basal friction coefficient, but not for ice viscosity at the same time which is actually pretty standard these days when using SSA (see for example Fürst et al. 2015, Cornford et al. 2015). This would also help to identify regions of shelf weakening in response to your pinning point being there (see Berger et al. (2016)) and also how these regions have changed as the size of the pinning point has changed.

Simply because inverting simultaneously for both (friction and rheology) at the same node would have the same effect on ISSMs cost function and therefore an infinite number of solutions - we did not find a way to implement a solution in ISSM to differentiate between grounded and floating nodes at the same time. As a possible solution, we follow Barnes et al. (2021) and Still and Hulbe (2021) and also carry out the inversion for each parameter independently. They also found that a consecutive inversion yields very similar results to *'whether or not the two inversions are performed simultaneously'* when compared to the Ua and STREAMICE ice-sheet models. This helped indeed to identify regions of soft rheology, particularly along the shear margin between the Eastern Ice Shelf and the Western Glacier Tongue and in the small shear zone upstream of the pinning point (as hypothesized byu the reviewers). We've added the figure as requested.

**RV1-1d** Please also add what kind of ice temperature you employ? Is your setup isothermal? What is your rate factor? All these things are needed for a complete description of you setup.

The depth-averaged rheological parameter is a function of temperature/fabric, which are both not explicitly implemented in the model. We've added this to the model description.

**RV1-2 Technical corrections:**
**RV1-2a Abstract:**
L5 I would change "model perturbation" to "ice-sheet model perturbations" to be more precise
Agreed and implemented

L43 ...emergent ice rise ... This could use a citation
Citation added

L72 Similar to L5, I would suggest to change it to "ice-sheet model perturbation experiments"
Agreed and implemented

[Figure]

L86 How come the Jordan et al. (2020) map has resolution of 1000 m, but you say in L82 that airborne gravimetry data cannot resolve anything below 5000 m?

The airborne gravimetry data was collected along flight lines of Operation IceBridge (2009) and the ITGC (2018/19). Here a figure and the methods from Jordan et al. (2020):

> *"All line data were then merged into a single database, interpolated onto a 1 km mesh raster, and filtered with a 5 km low-pass filter removing residual line-to-line noise. This filter wavelength is justified, as anomalies with wavelengths <5 km are not resolved by the airborne gravity systems used."*

So it just comes from gridding their data

*Illustration 5: from Jordan et al. (2020) showing the data acquisition lines from different sources to compile their gridded bathymetry*

L97-105 How do you ensure that you have a smooth transition (no jumps) between the region where you adjust the bathymetry and the background bathymetry?
We clip the extent of the 2011 pinning point from the adjustment plane and add it onto the published bathymetry. This way we create a 200m jump, which might introduce instabilities to our ice-sheet model experiments if the spatial extent of the pinning point would grow after 2011. As we have learned from the analysis of REMA and ICESat-2 data this is not the case and its extent shrinked since 2011. We hope that future bathymetry estimates from the inversion of gravity data will be constrained at this isolated location and therefore state our finding in the abstract. Text added.

L130 Equations: Here you write explicitly that w is a function of space. Either you do this everywhere (Eq. 4 and 5) or you drop it in all equations. Just be consistent. I think I would prefer the former but leave this to the authors to decide.
Thanks for pointing this out. We decided to remove the (x,y) part in the equations and use bolt fonts throughout the paper instead.

L135 I think it would really help to add a table with symbols/constants and values used in the study. Coming from ice dynamical modelling, the letter q is reserved for ice flux, so I think adding a table with all symbols and constants would be beneficial. You could then also drop the values in the main text.
Table added to the Appendix and values dropped in text where appropriate.

L155 Is there also an uncertainty associated with REMA?
The accuracy of REMA was estimated to be less than 1 m for the entire Antarctic continent (Howatt et al., 2019). However, the error estimate with the tile we are using on Thwaites is about 6 m, which largely originates from the mosaic seam outside of our area of interest. We've included this error a bit later in the paper in the discussion of the 1.25 m/yr accuracy of the derived surface lowering rates between 2014 and 2020.

L178 How do you get the uncertainty in grounding-line retreat rates? Does that not depend also on the geometry of the region (e.g. less inclined bed being more uncertain?)?
We manually picked the grounding-line products several times and calculated an average euclidean distance between the lines. This was about 300 m along a 150 km long grounding line, which corresponds to 0.1 km/yr over 3 years between each delineation. Yes, the true uncertainty will depend on the bed slope. Shallower sloping areas have a much higher accuracy than steeper sloping areas, because the grounding-line retreat will be more pronounced. Linking grounding-line retreat rates to bed slope would certainly be useful to assess whether there has been a change in the underlying mechanism/forcing (CDW intrusion vs dynamic thinning), but this is not exactly within the scope of this paper. We only present the new 2014 and 2020 grounding-line products to the community and would prefer to keep this analysis for future research.

L191 Again, can you briefly say how you arrive at these numbers?
We compare GPS measurements at 38 sites with the velocity fields from LandSAT-8 data in 2019 and find a mean difference of 28 m/yr, which corresponds to about 4.5 % of the average flow speed (624 m/yr). These numbers have changed slightly since our initial submission from a mean difference of 25 m/yr. We updated the numbers and added a few words that these numbers come from the GPS validation.

L213 Shouldn't that be "Shelfy"?
Thanks for spotting this typo.

L223 How can your mesh be 100 m resolution, if you are using mesh adaptation? I think you should move this to the end of the previous paragraph (L218) and give a max/min resolution range.

Agreed and implemented. This was initially our minimum resolution, but the word 'minimum' must have fallen through the cracks in the process. We added a sentence and a figure showing the mesh with boundary conditions to the supplementary material.

L224-225 I would delete the sentence starting with "We set grounded ...". You could set it to 1000 m/yr, but if you are running diagnostic experiments without geometry updates, this does not matter.
That's what we meant with 'instantaneous geometry' but agreed and sentence removed to avoid this confusion.

L230 I think here or somewhere nearby, it would be good if you could add what kind of boundary conditions you are applying in your experiments.
Dirichlet boundary conditions are imposed on the upstream boundaries of the computational domain using the ice thickness field and the two observed velocity field. Neumann boundary conditions are imposed along the ice front. Text added.

[Figure]

*Illustration 6: Model mesh and boundary conditions for a domain that includes the Glacier Tongue. The few edges on the pinning point that falsely show a Dirichlet Boundary condition are an artefact of the mesh-refinement procedure, where initially edges along the ice/water boundary now cut across the grounded portion of the pinning point.*

L254 How do you compute grounding-line retreat rates of ice rises? Is this a maximum estimate or a mean estimate? Would be good to add this.
We measured the minimum distance between the two remaining portions of the pinning point in 2011, 2014 and 2019/20. The 'retreat rate' here is consequently more a 'separation rate', or the rate of 'opening' between the two remaining portions. We have added a sentence.

L311 If I understand the reported data here correctly (8.0 HAF and 0.3 m/yr thinning), I find your formulation in the abstract and conclusions a bit too certain ("will unpin in less than a decade"). I think based on the extrapolation of your data it's more likely to happen within the next 20-30 years. So I would suggest to add a qualifier to these statements along the the lines "with the potential to detach within the next decade"

Agreed and implemented. However, unpinning is a non-linear process therefore we expect it to happen at the faster end of this range. Reworded according to IPCC convention (very likely).

L313-315 To interpret this we certainly need for information on regularisation etc. See main concerns.

Agreed, we now show the results of the L-curve analysis in the Appendix. Grounded friction values are now somewhere between 30-1000, but a lot smoother distributed than in our initial submission.

L320 What is "the isotropic consideration"? Do you mean that your rheology assumes that ice is isotropic?

Initially we employed a constant rheology throughout the domain (that's what we meant with isotropic consideration). However, we now also invert for ice rheology and discuss the results in the text.

L324 You need to say somewhere what stress model you use for you diagnostic simulations, but ideally before in section 2.5

Agreed, we now use SSA for both the inversion steps and the diagnostic step.

L325 Here you introduce the acronym ISSM, but you have used this already before. Please make sure that you introduce acronyms at first mention. I am not sure exactly what the TC policy is, but REMA and I believe MEaSUREs were never introduced. I am familiar with the data products, but I am not sure if journal policies require you to introduce these as well.

We have moved the indroduction of ISSM to the section 'Ice-dynamics modeling', REMA to the end of the introduction and MeaSUREs also to 'Ice-dynamics modeling'. Thanks for spotting this.

L336-337 Based on the evidence that you show, we know that the direction is correct. What about the magnitude? I think adding a 2D velocity magnitude difference plot would be helpful.

We have added a figure to the supplementary material showing the velocity mismatches for both control runs.

L337 When you remove the Western Glacier Tongue, do your boundary conditions change? What boundary condition do you apply where the Glacier Tongue was removed?

Along the upstream limit of the domain we impose a Dirichlet boundary condition, along the ice-shelf front we impose a Neumann boundary condition. So by changing the domain, the boundary conditions for grounded/floating ice do not change. When we remove the Glacier Tongue, grounded edges (according to our grounded/floating mask) along the new boundary automatically get the Dirichlet BC assigned, floating edges a Neumann BC.

[Figure]

*Illustration 7: Model mesh and boundary conditions for the domain after the Glacier Tongue has been removed.*

L348 Are you still using the same basal friction field or did you rerun the optimisation with the updated velocity dataset? If the former, please add a point of why you think this is justified?
We run the optimisation with the updated velocity data set (simulations c and d in Table 1). Thanks to the L-curve analysis we know that the inversion of the basal friction field isn't sensitive to what velocity data is used. Only the rheology changes, which is expected with a weakening shear margin. Control run means that we use this run to invert for basal friction and rheology and then use the derived fields to experiment with (I) changing the domain and (II) changing the grounded/ungrounded mask in the pinning point simulations.

L375 What are "fast horizontal gradients?" Do you mean large? I do not think gradients can be fast.
Agreed and implemented

L377 I just do not quite follow, how can reduced lateral drag result in slower ice velocities? Could you please explain this a bit more?
The Eastern Ice Shelf is dragged along by the much faster flowing Western Glacier Tongue. With a weakening shear margin, absolute lateral drag is reduced, which results in slower ice velocities in the East. Weakening of the shear margin in recent years was shown by Alley et al., 2021 (TCD, in review). What was once a strong shear margin, pushing solid ice towards the East, is now only a conglomerate of loosely attached icebergs in an ice melange.

L387 Why can it not be thicker ice that is being advected? Is there no evidence of thicker ice upstream?
Exactly, the airborne radar transects show that nicely. We added a reference to Figure 2.

L401 I would change to ...if no other forcings are changed such as ..."
Agreed and implemented

L416-417 The reported limits of the basal friction coefficient? I do not understand this sentence. Also as mentioned previously, you could incorporate anisotropic effects for the ice shelf into the model the same way you do for the basal friction coefficient, if you also invert for ice viscosity.

Although we now include an inversion for ice rheology, the inverted basal friction coefficients are still high, particularly at the pinning point and along the grounding line.

L432 I agree that it is always good to have multiple complementing methods. However, I feel that it should be mentioned somewhere that your flotation assumption is violated at the grounding line and hence will never give you the "true" grounding line location. Also I found that height above flotation is also quite sensitive to density estimates of the ice column and ocean.

We agree that differential InSAR is the most accurate method to delineate the grounding line. SAR data of sufficient temporal resolution to be coherent along Thwaites Glacier's fast-flowing grounding line are not acquired routinely (only through dedicated SAR data acquisition missions). Calculating height above flotation from two independent data sets (REMA and IceSAT-2) allowed us to provide the community with two new grounding-line mappings as well as to refine periods of rapid retreat seen in InSAR. Using two different ice density values for the inversion of freeboard to ice thickness has indeed the potential to bias the location of the grounding line. However, the density difference of 22 kg m-3 is too small to significantly move our grounding-line estimate beyond the pixel resolution and accuracy of manually delineating it. We reworded and added a sentence about the violation of the hydrostatic equilibrum assumption to the methods section.

L476 delete vigorous
Agreed and implemented

L488 I would add a qualifier here: "could reach flotation ..."
Agreed and implemented

**RV1-2b Figures:**

Fig. 2: Could you indicate ice flow direction also in panel (a). The dashed white line is also not visible in the legend
Agreed and implemented

[Figure]

*Illustration 8: Updated figure as suggested by both reviewers*

Fig. 3: How did you compute retreat rates? Along flowlines? Maximum retreat? Especially retreat for region IV in panel (c) looks strange. I am also not sure if panel (a) is needed. You could potentially delete this and change the layout to Fig. 5

The computed retreat rates are maximum and generally guided by flowlines, except AOI IV, where there would be hardly any grounding-line retreat along the main flowline, but since this '2017 peninsula' became a series of pinning points in 2020, we have chosen to pick the grounding-line retreat along a bathymetric low where the retreat occurred. Panel (a) is crucial because we delineate the 2014 grounding line with these data and we decided against adding the location of the profiles from panel (c) to this already busy panel.

Fig. 6: The letters U and D are hard to read. Can you change them to black font colour?

Agreed and implemented

[Figure]

*Illustration 9: Updated figure as suggested by the reviewer*

Fig. 7: I like these 3D Figures. But can you add either a flow direction arrow or indicate in the caption that 3D Figures are rotated by 90 degrees? Or if this is possible, have the same orientation as all your 2D maps?

We decided for 3D figures, because they highlight the depth of the trough near the pinning point (a critical area of warm water intrusion). The problem with rotating the figure to the same orientation as the birds-eye ones is that the Eastern Ice Shelf hides this trough. To assist the eye of the reader, we added ice flow arrows and mention the rotation in the figure caption.

Fig. 9: I can hardly see the red region. Since your colour scale includes red, please change to something different.

Agreed and implemented, that was truly hard to see.

Sincerely,
Clemens Schannwell,

On behalf of all co-authors thank you for your time and sharing your expertise. Christian Wild.

**References**

Berger, S., Favier, L., Drews, R., Derwael, J., and Pattyn, F. (2016). The control of an uncharted pinning point on the flow of an Antarctic ice shelf. Journal of Glaciology, 62(231), 37-45. doi:10.1017/jog.2016.7

Cornford, S. L., Martin, D. F., Payne, A. J., Ng, E. G., Le Brocq, A. M., Gladstone, R. M., Edwards, T. L., Shannon, S. R., Agosta, C., van den Broeke, M. R., Hellmer, H. H., Krinner, G., Ligtenberg, S. R. M., Timmermann, R., and Vaughan, D. G. (2015). Century-scale simulations of the response of the West Antarctic Ice Sheet to a warming climate, The Cryosphere, 9, 1579–1600, https://doi.org/10.5194/tc-9-1579-2015

Fürst, J. J., Durand, G., Gillet-Chaulet, F., Merino, N., Tavard, L., Mouginot, J., Gourmelen, N., and Gagliardini, O. (2015). Assimilation of Antarctic velocity observations provides evidence for uncharted pinning points, The Cryosphere, 9, 1427–1443, https://doi.org/10.5194/tc-9-1427-2015

**RV2 General comments**

Wild et al. study recent changes in ice dynamics of Thwaites Glacier in West Antarctica and investigates the impact of a subglacial pinning point under the eastern part of the Thwaites Ice Shelf. The authors use a suite of satellite imagery and ice-penetrating radar data to characterize changes over the last decade. A numerical ice sheet model is used to perform diagnostic (fixed-geometry) experiments to understand the ice dynamics associated with recent changes, and to assess the consequences of potential near-future ungrounding of the pinning point.

Thwaites Glacier is a major contributor to sea-level rise in West Antarctica, a region where there is potential for non-linear future mass loss. The underlying processes are not well constrained and the study is thus a timely contribution to a growing body of research on the glaciers in the Amundsen Sea, West Antarctica. The study clearly fits within the scope of TC and should be of high interest for the glaciological and broader scientific community.

The manuscript is well written with an overall coherent story. In general, the datasets and observational methods used are thoroughly described and the conclusions are supported by the analysed data and model experiments. Overall I think this is an exciting study with some unique datasets and interesting model experiments giving us an improved understanding of ice-shelf-glacier behavior at Thwaites.

The few major comments I have relate to the numerical model setup and description, as well as the need for a broader perspective on previous literature in the Introduction, and considering wider implications in Discussion. These comments should be addressed by the authors before publication in TC can be considered.

I should mention that my expertise is mainly ice dynamics and ice-sheet modelling, rather than remote sensing and radar glaciology. My comments below reflect this; the Editor should perhaps consider perspectives from a more empirical-minded scientist as well to get a balanced overall evaluation of the paper.

The reviewer is clearly another specialist in numerical ice-sheet modeling and has also understood the paper and its relevance to the glaciological community. Since many of reviewer #2s concerns largely overlap the comments from reviewer #1 we have shortened our reply where this is the case.

**RV2-1 Major comments**
**RV2-1a Model setup and description**
The following comments largely regards Section 2.5 Ice-dynamics modelling. While this section already contains useful information about the model, it is in my view not yet complete and does not allow the reader to fully understand your model setup and experiments. As it stands, this does somewhat erode the strength of your findings about the importance of the pinning points and the potential effects of future ungrounding. To alleviate this, I list below some concrete suggestions which I hope that the authors will consider.

Thank you very much for concrete suggestions how to improve the modeling part of this paper. We have tried to follow them as closely as possible.

First of all, it would be nice with a brief starting paragraph describing what the rationale is behind using a numerical ice sheet model. As it is written now, this section starts a bit abruptly with the reader not really knowing where we are going.

Agreed and implemented a statement about our reasoning to employ an ice-sheet model to understand both past and present changes and to predict likely scenarios of future change.

Regarding the friction law (Eq. 5), you should mention that you are assuming a linear Budd-type law, and that this is just one among several possible friction laws in the literature, and whether/how you think using other friction laws often used in Antarctica (e.g. a Schoof or Tsai law) would influence your results, if at all (this could also be done in Discussion).

Agreed and implemented, although we do NOT experiment with other possible friction laws in this study.

Need to mention what the assumption is for the effective pressure (perfect hydrological connection between the ocean and the grounded ice base), what the equation for N is (can be done in-text), and also what processes/factors this formulation of the effective pressure neglect.

Agreed and implemented as N is the effective pressure corresponding to the difference between cryostatic overburden pressure and water pressure at the ice base.

It is unclear to me why you use one ice-flow approximation for the inversion of the friction coefficient beta (2D SSA), and another for the stress balance calculation (3D Higher Order). Isn't the inversion for beta essentially a series of stress balance calculations? In my view, the cleanest and most consistent approach would be to use Blatter/Pattyn (Higher Order) for everything. I assume this wouldn't be too computationally demanding since you're only doing diagnostic experiments. If the authors have a very good reason for not using the same ice-flow approximation across all experiments, this needs to be stated clearly. And if you're going from 2D SSA to 3D Higher Order, how do you deal with the boundary conditions? Are you rerunning in 3D once you have your 2D inversion (and why?).

We now use SSA for both the inversion and the diagnostic experiments to be in accordance with the analysis of Barnes et al., 2021 and Still and Hulbe, 2021. We justify the SSA with the hypothesis of Schroeder et al., 2016 that much of Thwaites Glacier's bed is thawed and the overlaying ice mainly sliding on soft subglacial till.

The description of the model mesh needs to be more specific. You mention that you're using an anisotropic mesh refined near grounding lines and narrow shear zones. What exactly is the mesh resolution here, and what is the resolution elsewhere? If you're using a 3D Blatter/Pattyn approximation, how many vertical layers do you use? Are they uniformly vertically spaced, or do you have more finely spaced vertical layers towards the bed? See also Minor comments below.

We have added a sentence about the number of elements and their minimum/maximum spacing. Calculations are in 2D only with a depth-averaged rheology.

How is grounded ice rheology chosen/inferred? What ice temperature is assumed, based on what? How well constrained is this? What are associated uncertainties and potential influence on results?

This and the next comment. In a first inverse step, we now derive a rheology field using a suitable regularization parameter, w_{3}, in the cost function:

$$\mathscr{I}(\mathbf{p}) = I(\mathbf{p}) + R(\mathbf{p})$$

$$= w_1 I_{\mathrm{abs}}(\mathbf{p}) + w_2 I_{\mathrm{log}}(\mathbf{p}) + w_3 R(\mathbf{p})$$

$$= w_1 \int_S \frac{1}{2} \left[ \left( v_x - v_x^{\mathrm{obs}} \right)^2 + \left( v_y - v_y^{\mathrm{obs}} \right)^2 \right] dS +$$

$$w_2 \int_S \left[ \log \left( \frac{\|v\| + \epsilon}{\|v^{\mathrm{obs}}\| + \epsilon} \right) \right]^2 dS +$$

$$w_3 \int_\Omega \frac{1}{2} \|\nabla \mathbf{p}\|^2 \, d\Omega.$$

where p is firstly the depth-averaged rheological parameter B, and later the basal friction coefficient. The absolute and the logarithmic velocity misfit are equally weighed with w_{1} = w_{2} = 1 and have the same magnitude as the regularization term.

How is ice-shelf rheology chosen/inferred? What ice temperature is assumed, based on what? How well constrained is this? What are associated uncertainties and potential influence on results?
No temperature distribution is explicitly implemented, rather the rheology is still where ice is cold/strong. Text added

A reference to Table 1 in this section is missing; it would be great if you could briefly describe the experimental setup already here. As it stands, you have mostly outlined the model setup, while the details of your different experiments are missing in this section and are instead described in the Results (Section 3.4). One option is to move parts of Section 3.4 that concerns the experimental setup to Section 2.5.
Agreed and implemented

**RV2-1b Description and discussion of model results**
The following comments regard mainly Section 3.4 Numerical modeling of the ice-shelf response. In general this section is interesting and the experiments are potentially very revealing. However I'm missing key information regarding the model (see above) and the experimental setup. Therefore it is difficult to fully evaluate the insight gained by this modeling exercise.

You artificially increase basal drag to account for lowered ice viscosity in shear zones; is this straightforward or are there some pitfalls in doing this? Would it be possible to do two separate inversions, inverting for ice rheology (stiffness) for floating-ice areas, and for basal drag for grounded-ice areas? I assume this wouldn't be too much extra work, and wouldn't add much computation time either since you're only doing time-independent stress balance simulations. Adding variable ice-shelf rheology could move your model setup in a more realistic direction and potentially strengthen your interpretation of the importance of the shear margins on ice-shelf flow, and perhaps even reduce the need to increase the friction coefficient 'artificially' so much at the pinning point.
We now also invert for rheology and perform an L-curve analysis to find an appropriate regularization parameter.

A related question about the inversion and velocity calculations. How well do the modelled velocities actually fit with observed ones, for the 2009 and 2019 observations? What are the RMSEs? Do they consistently under/overestimate? Are there some problematic areas? What about grounded vs floating regions? I think this should be included in a supplementary figure, to give the reader an idea of the model

behavior. In particular, we're very interested how well the model captures your key regions of interest (the offshore pinning point, the grounding line, the shear zones). On L336-337 for example you state that "... the inversion is successful..." - more specifics would strengthen this section and further indicate whether your inferences about current and future ice dynamics are robust.

We have added a figure to the Supplementary Material that shows the model minus observation maps for the two different velocity data sets. Generally, the Eastern Ice Shelf and the pinning point are well matched by the model and parts of the Glacier Tongue, which have largely calved off, are missed.

I would like to applaud the authors for the nicely constructed experiments to test the influence of the breakup of the Western Glacier Tongue (Figure 8 and S4).

Thank you very much for the compliment regarding the design of our modeling experiments.

The 'ungrounding experiments' (Fig 9) are very interesting, but I would be a bit cautious, since your experiments are stress balance (diagnostic) simulations, and we do not know whether this increase in ice velocity will quickly dissipate or whether it will cause sustained ice speedup. Of course if the latter is the case, this will have major implications for the future of Thwaites over the next decades. You say in the Discussion that you see the breakup of the Western Glacier Tongue as an analog for future post-ungrounding of the Eastern Ice Shelf, and you clearly state in the text that you're indeed only doing diagnostic experiments. For balance, I think you should also state (in the abstract and/or conclusions) that model experiments with evolving geometry and ice flow would shed further light on the future dynamics and stability of Thwaites. To be clear, I think that transient experiments are out of scope of the current paper, and I do not expect the authors to include this in a revised version of the manuscript.

True, our experiments are only about observing instantaneous changes in the stress balance. Several other studies investigated the transient response of Thwaites Glacier to (I) a warming climate (Cornford et al., 2015), (II) sub ice-shelf melt rate (Joughin et al. 2014) and more recently (III) to ice-cliff instability (Bassis et al., 2021). All of these studies show that simulated ice loss/acceleration do not dissipate quickly. We added a statement to the conclusion as suggested by the reviewer.

**RV2-1c Wider implications**

I would like to see some more elaboration on previous work/state-of-the-art in the Introduction, and related to this, a broader discussion of the study's findings in the Discussion.

The Introduction overall reads well, with a good introduction to pinning points, ice rises and rumples. There is an appropriate background about Thwaites, including both changes over the long-term (Holocene) and the recent (last decades) past. However, as it stands, the Introduction is a little bit too Thwaites-heavy for my taste. There is scope for expansion about other glaciers than Thwaites, and why studying pinning points/ice shelf retreat is vital. There is relevant information about this already in the first two paragraphs, but the rest of the Introduction talks mostly about Thwaites. Of course Thwaites is a key glacier which the community strives to understand, but there are also others (in Antarctica, Greenland, and beyond). What are the particular knowledge gaps in our current understanding of pinning points/ice-shelf buttressing? What have others done before, and what does this study bring that is novel and different? I think by all means the data and findings in the study are novel, it's just that you can set the stage more clearly already in the Introduction, and then later pick up on this in the Discussion.

We agree that studying pinning points around Antarctica is vital and have therefore included an elaborate overview and classification in the introduction. We are convinced that (at least some) of the techniques, which we developed for this study, can not only be applied to Thwaites Glacier but also to other pinning points. The present study, however, is intended to specifically target Thwaites pinning point and we feel that 'widening' the discussion section beyond Thwaites would unnecessarily prolong an already lengthy paper. The novelty of this study is highlighted by discussing the contradiction with the modeling results from Nias et al., 2016 in a dedicated paragraph. Furthermore, we provide observational evidence of upstream thickening – a process described by Still et al., 2019.

**RV2-2 Minor and technical comments**

In general, more cross-references to figures and other sections would be great to help the reader.
We have added cross-references where appropriate.

**RV2-2a Abstract**
L5. "model perturbation experiments" – could be more specific of what kind of model (ice-shelf-glacier model, numerical model, etc)
Reworded to 'ice-sheet model perturbation experiments'.

**RV2-2b Introduction**
L51. retreat rates -> please specify if you talk about retreat rates of the grounding line or the ice-shelf front here (I think you mean the former)
Added 'grounding-line'.

**RV2-2c Data and Methods**
L86-87. Not entirely clear on the use of bed/bathymetry data here. Please clarify which dataset you use where; how are they integrated/combined? (Jordan et al., 2020 and Morlighem et al., 2020).
Both bathymetry data sets are compared individually to the airborne radar transects across the pinning point (and not integrated). We reworded this sentence to make this more clear.

L92-93. Would be nice to add a cross-reference to Fig 1a here
Agreed and implemented.

L95-96. This is useful information – please specify why (mechanisms/processes) a crevassed (smooth) ice base indicates that ice is in disequilibrium (equilibrium) with the underlying ocean.
We are actually preparing an entire paper on this topic and would therefore prefer to only mention this disequilibrium here. These processes are (I) a small-scale violation of hydrostatic equilibrium, which we also indicate in Figure 6 and describe in the text as a series of 'alternating thickening and thinning signals'. And (II) heterogeneous basal melt patterns induced by accelerated melting on almost vertical ice faces on the ice base.

L168. Which observation? Please clarify.
Mean ice-column density. Text added.

L182-183. Typo; should be "Lagrangian"
Agreed and implemented. Thanks for pointing out this typo.

L199. Could you be specific on how large this contribution from REMA is, compared to the others mentioned on L196-197?
We calculate surface lowering rates between 2014 and 2020. REMA has an absolute vertical error of less than 1 m (Howatt et al., 2019), although the area-wide average in the provided error tile is about 6 m. ICESat-2 has an accuracy better than 3 cm (Brunt et al., 2019). The tide correction from comparison of the CATS tide model with two freely-floating GPS receivers on the AMIGOS  has +- 17cm. In the tidal flexure zone, inaccuracies from the value of the Young's modulus translate to +-0.08 m. If we take the most conservative estimate and add all of these components, divided by the 5 to 6 years, the inaccuracy is up to about 1.25 m/yr, which is well below the discussed signals that feature several m/yr. Text added.

L203. should mention what the acronym ISSM stands for, as well as add reference (Larour et al., 2012?)
Agreed and implemented.

L213. Typo: should be "Shelfy-"
Agreed and implemented. Thanks for pointing out this typo.

L218-219. "... guided by surface velocity fields..." – this is a bit vague, please be more specific how you construct the model mesh. Do you only use the gradient of the observed surface velocity fields (which one), or also set the mesh resolution manually in some areas? You mention on L223 that the horizontal resolution of the final mesh is 100 m; does this mean that your mesh resolution is uniform over the entire domain?

We added a more detailed description of our finite-element mesh and a Supplementary Figure:

[Figure]

*Illustration 10: New figure showing the model mesh as described in the text*

**RV2-2d** Results**

L232. Figure 3d is not discussed explicitly in this section; perhaps it should be at least cross-referenced to point the reader to this useful summary of temporal evolution of observed retreat rates.
We added a sentence.

L240-263. What would improve the dynamical insight of this section further is to link these observed changes in grounding-line locations to changes in surface elevation or ice thickness (for example those shown in Figure 5c, but also elsewhere on the larger ice-shelf and upstream of the grounding line).
We agree and thank the reviewer for the suggestion. As outlined above, linking grounding-line retreat rates to other variables such as a DEM, ice thickness or bed slope is beyond the scope of this paper and we would like to keep this analysis for future research.

L265-292. Overall nicely written and interesting findings. Could mention and add cross-references to modeling results (section 3.4) to guide the reader further through this section.
We've added references to the modeling figures where appropriate.

L305. Please explain what a sastrugi is for readers not familiar with snow dynamics.
We added a few words about their origin.

L308-311. Your final point about ungrounding of the pinning point is important as it is one of the main conclusions of the paper, and even occurs in the title. From the numbers you have provided (avg. height above flotation 8.0 +/- 6.5m and avg. surface-lowering rates 2014-2019/2020 of -0.3 +/- 1.5 m/a), I think you should be a bit more careful with stating that the pinning point may unground in less than one decade. Yes this is possible if the true height above flotation is on the lower end of your stated 8.0 +/- 6.5m, and/or if the true surface-lowering rates are on the higher end of your stated -0.3 +/- 1.5 m/a. Unless I misunderstood something, if you take the average values 8.0 m and -0.3 m/a, complete ungrounding will take 26.7 years. This is of course still interesting and potentially somewhat alarming, but not as dramatic as "less than a decade". If my understanding of what you have done is correct, I think you can phrase this in a more balanced way (as well as in the abstract and conclusions); you should indeed mention what "could" happen, while explaining what is the "best-guess" (average?). I think you can also state more explicitly here (as well as in the abstract and/or in the conclusions) that your inferred timing of ungrounding is merely an extrapolation of recent thinning rates into the future, and not a prediction. Running a numerical model forward in time could obviously shed light on this in more detail.
We follow your suggestion and have implemented several changes regarding the wording. However, we still believe that the ungrounding will happen at the sooner end of the spectrum.

L313-315. what is the minimum friction coefficient allowed in the inversion, is it 10 (Pa yr/m)1/2 ?
It is 1 (Pa yr/m)1/2. The maximum 1000 (Pa yr/m)1/2

L334. When inferring basal friction coefficients, why do you use the 2011 grounding line for both the MEaSUREs 2009 (Figure S5a) and Landsat-8 2019 (Figure S5b) velocity fields? Why not use the updated grounding lines from 2019/2020? You state on L334 that all simulations use the 2011 grounding line. Please explain what the reasoning behind this is.
We target the effect of weakening the pinning point on the regional ice dynamics/grounding-line discharge and have therefore kept the main grounding line the same reference for all simulations. Implementing two different grounded areas on the mainland would be possible in theory, but would also complicate the interpretation of our results. The reason is that there is only one ice thickness field available for both simulations and moving the grounding line inland would spuriously increase ice discharge across the grounding line by a thickening ice columnn. By keeping the same grounding line, we can make the statement that grounding-line discharge increases with weakening/removal of the pinning point.

L358. I think you mean "model elements" rather than "pixels" ?
Agreed and implemented. Exactly, thanks for pointing this out. Guess that came from months of remote sensing.

**RV2-2e Discussion**
Overall very well written, discusses main sources of uncertainty and suggests next steps for improvement, as well as discussing results in light of previous studies. A wider-scope discussion related to other Antarctic glaciers and implications for our understanding of ice-shelf buttressing and ice-shelf dynamics in general would improve this section even further (related to my comments about the Introduction), see also Major comments above.
Thank you very much for the positive feedback regarding the discussion of our results.

**RV2-2f Conclusion**
L485. I think you should state what kind of model, and that simulations are fixed-geometry velocity snapshots, and thus does not establish whether this will be a short-lived or sustained feature
Text added as suggested.

L493. "... possibly the most likely" – this seems a bit vague or even contradictory. And based on what exactly? You could be more specific here.
We like to follow the wording convention of the latest IPCC report used to indicate the assessed likelihood of an outcome or a result (i.e. virtually certain, very likely, likely,…). We rearranged the sentence and reworded therefore to 'very likely'

**RV2-2g Figures**
In general, figures are nicely crafted and clear.
Thank you, they were a lot of fun to put together

Figure 1

Missing information on what A and B means in (a) – I think they are airborne radar flight lines? Also, in (a) and (c), the colorbar labels should be m above sea level (m a.s.l.)
Agreed and implemented. We added A and B to the caption with a reference to the radargrams.

[Figure]

*Illustration 11: Updated figure as suggested by the reviewer*

Figure 2
The dashed white line for hydrostatic equilibrium is not visible in the legend
Agreed and implemented. It's light blue now (see illustration above)

Figure 7.
Would be useful if these figures had the same orientation as Figures 1-6 and 8. This also applies for Figure 9.
We have experimented with the orientations, but the reason why we would like to keep the given orientation of the 3D figures is that they highlight the depth of the marine trough in comparison to the height of the pinning point (almost a 1500m vertical drop !). If we rotate to the suggested orientation, the Eastern Ice Shelf would block this view. To assisst the reader, we have added flow direction arrows and mention the rotation now in the caption.

Figure S1. Missing axis labels and units in all panels.
This figure is more qualitative to guide the reader through the adjustment process. We have therefore removed the axes ticklabels to lighten this already busy figure. The unit of all displayed data is m. We added this to the caption.

Figure S4a. In the legend, "ice (m)" is a bit cryptic. I think you mean "ice thickness" ? In the supplementary, I would also like to see plots of the modelled velocity magnitudes and their mismatches with observed velocity fields (2009 and 2019), not only the velocity directions (see comments above).
Agreed and implemented.

Thank you very much for your comments and sharing your insight into ice-sheet modelling.

**References cited in response to both reviewers:**

Alley, K. E., Wild, C. T., Luckman, A., Scambos, T. A., Truffer, M., Pettit, E. C., Muto, A., Wallin, B., Klinger, M., Sutterley, T., Child, S. F., Hulen, C., Lenaerts, J. T. M., Maclennan, M., Keenan, E., and Dunmire, D.: Two decades of dynamic change and progressive destabilization on the Thwaites Eastern Ice Shelf, The Cryosphere Discuss. [preprint], https://doi.org/10.5194/tc-2021-76, in review, 2021.

Barnes, J. M., Dias dos Santos, T., Goldberg, D., Gudmundsson, G. H., Morlighem, M., and De Rydt, J.: The transferability of adjoint inversion products between different ice flow models, The Cryosphere, 15, 1975–2000, https://doi.org/10.5194/tc-15-1975-2021, 2021.

Brunt, K. M., Neumann, T. A., & Smith, B. E. (2019). Assessment of ICESat-2 ice sheet surface heights, based on comparisons over the interior of the Antarctic ice sheet. *Geophysical Research Letters*, 46, 13072– 13078. https://doi.org/10.1029/2019GL084886

Howat, I. M., Porter, C., Smith, B. E., Noh, M.-J., and Morin, P.: The Reference Elevation Model of Antarctica, The Cryosphere, 13, 665–674, https://doi.org/10.5194/tc-13-665-2019, 2019.

NIAS, I., CORNFORD, S., & PAYNE, A. (2016). Contrasting the modelled sensitivity of the Amundsen Sea Embayment ice streams. *Journal of Glaciology, 62*(233), 552-562. doi:10.1017/jog.2016.40

Still, H., Campbell, A., & Hulbe, C. (2019). Mechanical analysis of pinning points in the Ross Ice Shelf, Antarctica. *Annals of Glaciology, 60*(78), 32-41. doi:10.1017/aog.2018.31

Still, H. and Hulbe, C.: Mechanics and dynamics of pinning points on the Shirase Coast, West Antarctica, The Cryosphere, 15, 2647–2665, https://doi.org/10.5194/tc-15-2647-2021, 2021.

---

## Referee Report (RR1)

**Review of tc-2021-130, revised version**

Dear Christian Wild with co-authors,

I have now gone through the revised manuscript and read your responses to the initial comments from reviewer 1 and 2 (myself). I must say that the authors have done a great job addressing the comments raised by both reviewers. This has turned out as an excellent paper. However, a few aspects remain, and I am going to suggest that some minor revisions are needed.

There a few things that should be addressed, outlined below. The line numbers refers to the tracked-changes document, tc-2021-130-ATC1.

**HO vs SSA ice flow**

Great that you now are using the same ice-flow approximation across the inversions and the forward run. I think using SSA for all is ok; however I do see that there are some differences between how well HO and SSA represents the observed velocity field, as shown in Illustration 3 in your response to reviewers. The continuous colormap for the difference maps in Illustration 3 (right-hand side panels) makes it a bit hard to compare the model – observed misfits (a red/white/blue difference colormap would have been better). I understand that you'd like to save a bit of computational time by using SSA for all, and I am 100% supporting you to stick to the same ice-flow approximation across all experiments. However I am tempted to suggest that you should use HO for everything, both because you have it so readily available in ISSM and also because you have already done half of the job. It would not (?) be so much extra work to use HO for the inversions as well, as they are all stress balance calculations, in contrast to the forward transient runs. I am aware however that the manuscript is at an advanced stage. Alternatively, the authors need to clearly justify why HO is not needed, and illustrate that the results and conclusions are essentially the same as with SSA only.

**Minor/technical comments**

L210. A very minor comment: I would write "recent" rather than "past" to distinguish from more historical/paleo-type studies.

L223-24. ISSM uses a Budd-type friction law (Budd et al. 1984), not Weertman, as you have written now. Please rewrite. The equation 6 is still correct though. In my initial comment, I wrote that

*"Regarding the friction law (Eq. 5), you should mention that you are assuming a linear Budd-type law, and that this is just one among several possible friction laws in the literature, and whether/how you think using other friction laws often used in Antarctica (e.g. a Schoof or Tsai law) would influence your results, if at all (this could also be done in Discussion)."*

I cannot find where you have addressed the latter point (how using other laws would influence your results).

L226. Great that you have now added the equation for the effective pressure N. However, you have not yet addressed the first and the latter part of my initial comment. A sentence or two would be sufficient.

*"Need to mention what the assumption is for the effective pressure (perfect hydrological connection between the ocean and the grounded ice base), what the equation for N is (can be done in-text), and also what processes/factors this formulation of the effective pressure neglect."*

L250-52. Please explicitly state (in parentheses) what a Dirichlet and Neumann boundary condition means in practice here, to help the non-modelers among the readers.

L553-55. Thanks for rephrasing in a clearer, IPCC-like way. Perhaps I am being picky, but I think that the "very likely" ungrounding is still an hypothesis; I would be careful here. I'm not sure that "very likely" is consistent with the new, more balanced phrasing in L347-50. If you postulate ungrounding within the next decade (which I interpret as an anticipated acceleration of current thinning rates?), I would explicitly state why that is "very likely" to occur on the shorter time scale. What processes/feedbacks makes it "very likely" that current thinning will accelerate and ungrounding will occur sooner than later?

**Figures**
Figure 3b. Colorbar label not complete, m what?

Figure 7. (a). colorbar label should be ice surface (m a.s.l.); (b). colobar label should be bed elevation (m a.s.l.)

---

## Author Response (AR2)

**Author reply to both reviewer comments of tc-2021-130, revised version by Christian Wild (20 Dec'21)**

**Reviewer #1 : Clemens Schannwell**

Thank you for the second round of comments (all agreed) and for your very constructive review of our paper. We have thoroughly enjoyed this learning process and are excited about its outcome.

line 16: delete 'unprecedented' before ice-shelf thinning, agreed and implemented

Fig 1 caption: typo, agreed and implemented

Fig 2 caption: black arrows disappeared pointing towards areas with surface elevation above flotation height, agreed and implemented

Eq (4) and beyond: bold characters for vector/matrices/tensor only according to TC's policy: We changed throughout the manuscript following the author's guidelines.

From the copernicus Latex template for TC:

%%% Physical quantities/variables are typeset in italic font (*t* for time, *T* for Temperature) %%% Indices which are not defined are typeset in italic font (*x*, *y*, *z*, *a*, *b*, *c*) %%% Items/objects which are defined are typeset in roman font (Car A, Car B) %%% Descriptions/specifications which are defined by itself are typeset in roman font (abs, rel, ref, tot, net, ice) %%% Abbreviations from 2 letters are typeset in roman font (RH, LAI) %%% Vectors are identified in bold italic font using \vec{x} %%% Matrices are identified in bold roman font %%% Multiplication signs are typeset using the LaTeX commands \times (for vector products, grids, and exponential notations) or \cdot %%% The character \* should not be applied as multiplication sign

For example, ice thickness H is a 2d matrix and therefore non-italic and bold font, while basal velocity v\_b is a 2d vector and therefore italic and bold font.

Fig 3 caption: delete 'light' before blue, agreed and implemented

line 225-226: change start of sentence to 'Here we use the SSA to infer both', agreed and implemented, deleted 'Full-Stokes'

line 291: delete space between '3 d', agreed and implemented

line 355: reword 'swamp-like' to 'large patches of slippery subglacial conditions', agreed and implemented

Fig. 7: missing note about view rotation of the Figure, agreed and implemented

line 512: reword 'stress-balance experiments' to 'diagnostic experiments', agreed and implemented

**Reviewer #2 : Anonymous**

We thank the reviewer for providing insightful comments throughout the review process, especially regarding the thought provoking discussion about friction laws and assumptions surrounding the effective pressure.

**HO vs SSA**

We agree that using HO for both the inversions and the forward runs would be a further step in the analysis, but prefer to keep the SSA throughout the present study to be in line with the analysis of Barnes et al., 2021 for glaciers draining into the Amundsen Sea. The authors demonstrate that inversion products, particularly basal friction fields, are directly transferable between different ice flow models and their underlying stress-balance equations. When modeling an ice shelf and other areas of relatively low friction, then the SSA is the appropriate approximation to use. Higher order models would clearly do a better job in the area of transitional flow, especially the grounding line. But that is not the focus of the paper. It is possible that our grounding line flux estimates would change slightly. But given other uncertainties, such as temperature distribution this is not likely to be a major problem. We therefore believe that replacing the SSA with HO in our analysis would not be a major cause of differences nor impact the conclusions. Furthermore, the SSA has been used successfully in related perturbation experiments using ISSM by Still and Hulbe, 2021 for pinning points on the Ross Ice Shelf.

**Minor/technical comments**

line 210: replace 'recent' with 'past', agreed and implemented

line 223-24: replace 'Weertman' with 'Budd'-type friction law but keep the equation, agreed and implemented. We added a few sentences about alternative friction laws to the discussion. Budd includes the effective pressure, N, in the equation and Weertman depends on the friction coefficient only. Because N is a constant, an inversion without it would simply raise the value of the determined basal friction coefficient. A Schoof-type sliding law is treating flow over an undulated bedrock and becomes essentially plastic at high water pressure, just like Coulomb laws for till. The important feature of the basal sliding law is that friction becomes low for low N. That effect is captured in Budd, Schoof, Coulomb.

line 226: add a sentence about the assumptions regarding effective pressure and what processes this formulation neglects, agreed and implemented.

line 250-52: state in parentheses what a Dirichlet and Neumann boundary condition means in practice here, agreed and implemented

line 553-55: What processes/feedbacks make it 'very likely' that current thinning will accelerate. We discuss the nonlinear processes and our reasoning to state ungrounding occurs sooner than later within the reported range in the last three paragraphs of the Discussion section. We therefore only enumerate these again in the Conclusion. Text added.

**Figures**

Fig. 3B: Colorbar label 'm' what ? We changed to 'm a.f.' to indicate meters above flotation here and in the text

Fig 7: (a) label should be ice surface, disagreed, the displayed variable is ice thickness, which is in 'm' (b) label should be bed elevation in 'm a.s.l.', agreed and implemented